# CDK5-dependent phosphorylation and nuclear translocation of TRIM59 promotes macroH2A1 ubiquitination and tumorigenicity

Youzhou Sang [1,4], Yanxin Li [2,4], Yingwen Zhang [2], Angel A. Alvarez [3], Bo Yu [1], Weiwei Zhang [1], Bo Hu [3], Shi-Yuan Cheng [3] & Haizhong Feng [1]

Despite the development of adjuvant therapies, glioblastoma (GBM) patients remain incurable, thus justifying the urgent need of new therapies. CDK5 plays a critical role in GBM and is a potential target for GBM. However, the mechanism by which CDK5 promotes GBM tumorigenicity remains largely unknown. Here, we identify TRIM59 as a substrate of CDK5. EGFR-activated CDK5 directly binds to and phosphorylates TRIM59, a ubiquitin ligase at serine 308, which recruits PIN1 for *cis–trans* isomerization of TRIM59, leading to TRIM59 binding to importin α5 and nuclear translocation. Nuclear TRIM59 induces ubiquitination and degradation of the tumor suppressive histone variant macroH2A1, leading to enhanced STAT3 signaling activation and tumorigenicity. These findings are confirmed by inhibition of CDK5-activated TRIM59 activity that results in suppression of intracranial tumor growth. Correlative expressions of the components of this pathway are clinically prognostic. Our findings suggest targeting CDK5/TRIM59 signaling axis as a putative strategy for treating GBM.

[1] State Key Laboratory of Oncogenes and Related Genes, Renji-Med X Clinical Stem Cell Research Center, Ren Ji Hospital, Shanghai Cancer Institute, School of Medicine, Shanghai Jiao Tong University, 200127 Shanghai, China. [2] Key Laboratory of Pediatric Hematology and Oncology Ministry of Health, Pediatric Translational Medicine Institute, Shanghai Children's Medical Center, School of Medicine, Shanghai Jiao Tong University, 200127 Shanghai, China. [3] The Ken and Ruth Davee Department of Neurology, Lou & Jean Malnati Brain Tumor Institute, The Robert H. Lurie Comprehensive Cancer Center, Northwestern University Feinberg School of Medicine, Chicago, IL 60611, USA. [4] These authors contributed equally: Youzhou Sang, Yanxin Li. Correspondence and requests for materials should be addressed to H.F. (email: fenghaizhong@sjtu.edu.cn)

Glioblastoma (GBM) is the most deadly primary brain tumor with high heterogeneity and poor prognosis[1–4]. Despite the development of adjuvant therapies, most patients with GBM live for only 14 to 16 months after diagnosis, underscoring the urgent need for new therapies to combat this deadly cancer[1]. Over the past decade, cumulative evidence has demonstrated frequent protein kinase dysregulations in human cancers including GBM and critical roles of various kinases during tumor initiation and progression[5]. Thus, protein kinases are established therapeutic targets for cancer treatments[5,6]. However, clinically approved protein kinase-targeted therapies for GBM are still lacking.

Cyclin-dependent kinase 5 (CDK5) is an unconventional CDK that regulates neurogenesis rather than cell cycle[7] and is essential in central nervous system development[8,9]. CDK5 is highly expressed in the brain, and its activity depends on two coactivators, p35 and/or p39[10]. Activation of CDK5 is critical for angiogenesis, apoptosis, myogenesis, vesicular transport, and senescence in nonneuronal cells, including tumors[9,11,12]. CDK5 has been implicated in the pathology of multiple types of cancers[11,13,14] and is emerging as a potential therapeutic target for GBM[15,16]. In GBM, CDK5 is highly expressed in clinical tumors[17] and a critical regulator of GBM tumorigenesis[16]. Targeting CDK5 promotes survival of *Drosophila* with brain tumors[16], enhances antitumor immunity by reducing tumor PD-L1 expression[15], and impairs mitochondrial dynamics in brain tumor-initiating cells[18]. However, the mechanism by which CDK5 regulates GBM tumorigenicity remains unclarified.

Tripartite motif-containing 59 (TRIM59) functions as a ubiquitination ligase[19,20] or an adaptor protein[21,22], and plays important roles in various types of human cancers[19,23,24]. Recent studies from our laboratory demonstrated that TRIM59 promotes nuclear signal transducer and activator of transcription 3 (STAT3) signaling activity by inhibiting T cell protein tyrosine phosphatase (TC45) dephosphorylation in GBM[21]. In this study, we report a critical role of TRIM59 as a substrate of CDK5 in epidermal growth factor receptor (EGFR)-driven GBM. EGFR-activated CDK5 phosphorylated TRIM59, resulting in TRIM59 nuclear translocation via peptidyl-prolyl *cis–trans* isomerase protein interacting with NIMA (never in mitosis A) 1 (PIN1)/importin α5 axis. Nuclear TRIM59 then promotes the tumor-suppressive histone variant macroH2A1 ubiquitination and degradation, leading to enhanced STAT3 signaling activation and tumorigenicity.

## Results

**EGFR-activated CDK5 promotes TRIM59 nuclear translocation**. We first determined whether EGF induced TRIM59 nuclear translocation in GBM LN229 cells expressing exogenous wild-type (WT) EGFR. EGF stimulation significantly promoted TRIM59 nuclear translocation, whereas treatment with EGFR tyrosine kinase inhibitor erlotinib markedly reduced EGF-induced TRIM59 nuclear translocation (Fig. 1a, b). We then performed in silico analysis through NetNES 1.1 Server [http://www.cbs.dtu.dk/services/NetNES/][25] and identified a nuclear export signal sequence (233-LELMALTISLQEE-245) in TRIM59. Pre-treatment with leptomycin B inhibitor of nuclear export retained EGF-induced TRIM59 nuclear localization, which was inhibited by erlotinib (Fig. 1a, b). Additionally, LN229 cells overexpressing exogenous EGFRvIII, a constitutively active EGFR mutant that is frequently found in clinical GBM tumors and drives glioma tumorigenicity[2], had a higher amount of nuclear TRIM59 proteins than parental LN229 cells (Supplementary Fig. 1a, b). Recently, we reported that EGFR activation transcriptionally upregulates *TRIM59* mRNA levels via SOX9[21]. To further investigate whether

EGF stimulation affects TRIM59 protein stability, we treated LN229/EGFR cells with cycloheximide (CHX) after EGF stimulation and found that TRIM59 degradation was not affected by EGF treatment compared with the controls (Supplementary Fig. 2a, b). These results suggest that EGFR activation promotes nuclear translocation of TRIM59.

Next, we sought to identify which kinases are responsible for EGF-induced nuclear translocation of TRIM59. LN229/EGFR GBM cells were treated with or without PI3K inhibitor LY290042, MEK inhibitor U0126, Src inhibitor SU6656, CDK5 inhibitor Roscovitine, c-Jun inhibitor SP600125, CaMKII inhibitor KN-93, PKC inhibitor GF109203X, GSK-3 inhibitor SB216763, or PKA inhibitor H-89 before EGF stimulation, respectively. As shown in Fig. 1c, only treatment with CDK5 inhibitor Roscovitine markedly impaired EGF-induced nuclear accumulation of TRIM59 compared with the EGF-stimulated control. This observation was further validated by immunofluorescence (IF) (Fig. 1d) and western blot (WB) analyses (Fig. 1e).

To further determine the effect of CDK5 on TRIM59 nuclear accumulation, we knocked out CDK5 in LN229/EGFR cells using the CRISPR/Cas9 technology. CDK5 depletion markedly abrogated EGF-stimulated TRIM59 nuclear translocation (Fig. 1f). Moreover, re-expression of CDK5 WT, but not a kinase-dead (KD) mutant[26], restored nuclear translocation of TRIM59 in GBM cells (Fig. 1g). These data indicate that CDK5 kinase activity is required for EGF-induced nuclear translocation of TRIM59. To exclude possibilities that other relevant CDKs, including CDK1, CDK2, and CDK4, might also involve in CDK-mediated TRIM59 nuclear translocation, we knocked out CDK1, CDK2, and CDK4, as well as CDK5, in LN229/EGFR cells and assessed EGF induction of TRIM59 nuclear translocation. As shown in Supplementary Fig. 3, only CDK5 knockout but not other CDK knockout markedly reduced EGF-induced TRIM59 nuclear translocation.

**CDK5 directly binds to and phosphorylates TRIM59**. To determine whether CDK5 directly interacts with TRIM59, we first carried out an immunoprecipitation (IP) assay and found that endogenous CDK5 was associated with TRIM59 in LN229 and U87 cells expressing exogenous EGFRvIII (Fig. 2a). EGF stimulation also significantly enhanced CDK5 association with TRIM59 in EGFR-expressing LN229 and U87 cells (Fig. 2b). To further validate this observation, we performed glutathione *S*-transferase (GST) pull-down analysis and found that purified recombinant CDK5 directly interacted with TRIM59, and their binding was significantly enhanced by EGF stimulation (Fig. 2c). Additionally, ectopic expression of TRIM59 enhanced its association with CDK5 in a dose-dependent manner (Supplementary Fig. 4a). Lastly, we investigated the association of CDK5 WT or KD with TRIM59 WT, S308A, or S308D, and found that CDK5 KD or TRIM59 S308A mutant had reduced its ability to interact with their corresponding WT partner, CDK5 or TRIM59, respectively, suggesting that both CDK5 kinase activity and p-S308 of TRIM59 are critical for CDK5–TRIM59 interaction (Supplementary Fig. 4b). Moreover, when Flag-tagged CDK5 was separately co-expressed with TRIM59-truncated mutants, D1, D2, D3, D4, and D5 (Fig. 2d) in LN229/EGFR cells, EGF treatment promoted the D1 and D5, but not other D mutants binding with CDK5, suggesting that the C-terminal fragment (amino acids (AAs) 281–403) of TRIM59 is required for its association with CDK5 (Fig. 2d, e).

To determine whether TRIM59 is a substrate of CDK5, we assessed the effect of CDK5 knockout on TRIM59 phosphorylation. As shown in Fig. 2f, CDK5 knockout impaired EGF-stimulated TRIM59 phosphoserine/threonine (p-Ser/Tht) in

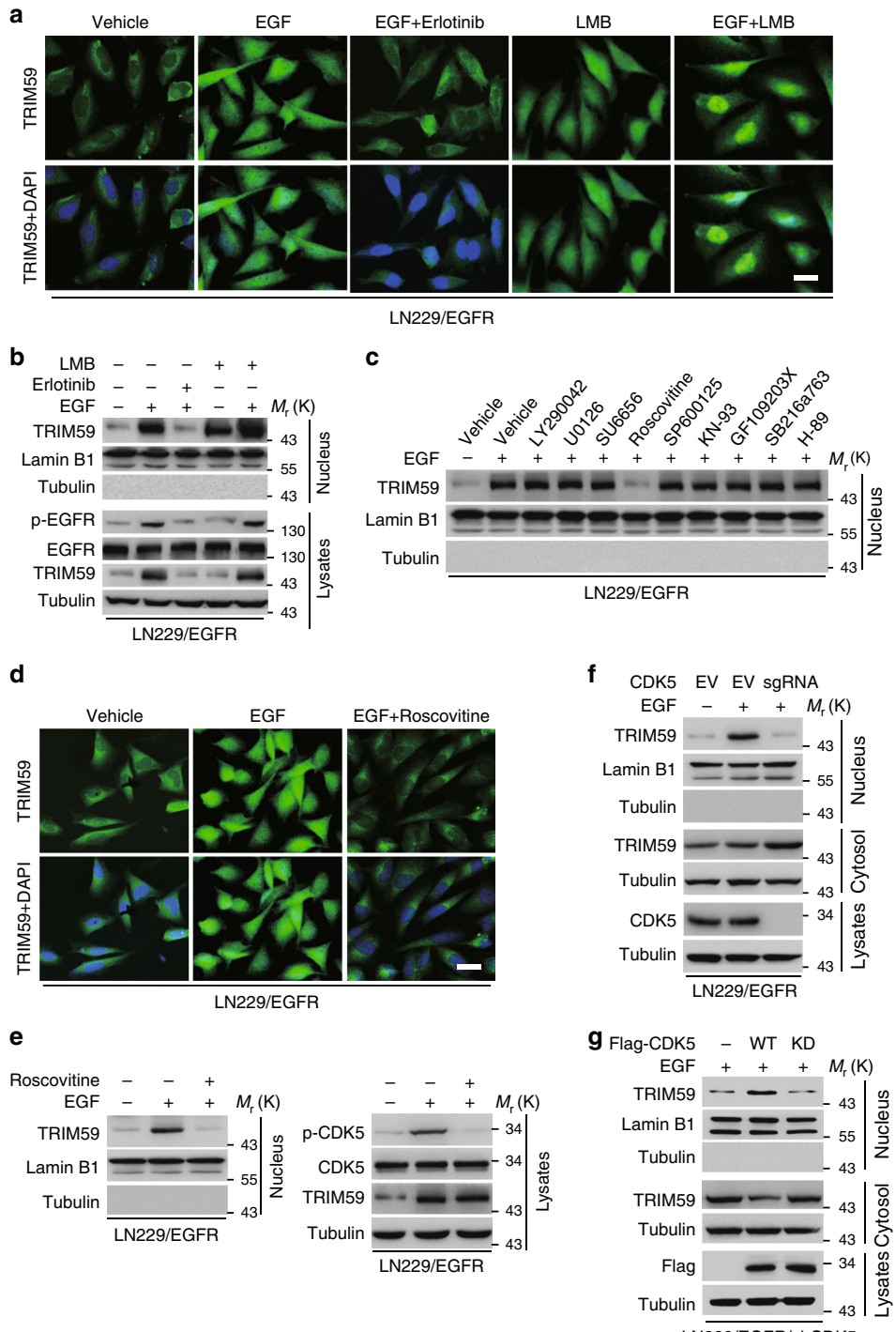

**Fig. 1** EGFR-activated CDK5 promotes TRIM59 nuclear translocation. **a** Representative images of immunofluorescence (IF) analysis of TRIM59 nuclear translocation stimulated by EGF. LN229/EGFR cells were pre-treated with or without erotinib (10 μM) or leptomycin (LMB, 20 nM) for 2 h, and then stimulated with EGF (100 ng/ml) for 6 h. Scale bar, 40 μm. **b** Western blotting (WB) for EGF-stimulated TRIM59 nuclear translocation. Nuclear fractions were prepared from LN229/EGFR cells in **a**. Nuclear lamin B1 and cytoplasmic tubulin were used as controls. **c** Effects of various kinase inhibitor treatments on TRIM59 nuclear translocation. LN229/EGFR cells were pre-treated with or without LY290042 (30 μM), U0126 (20 μM), SU6656 (10 μM), Roscovitine (20 μM), SP600125 (25 μM), KN-93 (250 μM), GF109203X (10 μM), SB216763 (10 μM), or H-89 (20 μM) for 1 h, and then stimulated with EGF (100 ng/ml) for 6 h. **d** CDK5 inhibitor Roscovitine inhibits EGF-stimulated TRIM59 nuclear translocation in LN229/EGFR cells. Scale bar, 40 μm. **e** Representative images of EGF-stimulated TRIM59 nuclear translocation; lower, WB analysis. **f** Effect of CDK5 knockout using a sgRNA on TRIM59 nuclear localization in LN229/EGFR cells with or without EGF stimulation. **g** Effects of reconstituted Flag-tagged CDK5 wild-type (WT) or kinase-dead (KD) mutant on TRIM59 nuclear translocation in LN229/EGFR/shCDK5 cells. Data are representative of three independent experiments with similar results. Source data are provided as a Source Data file

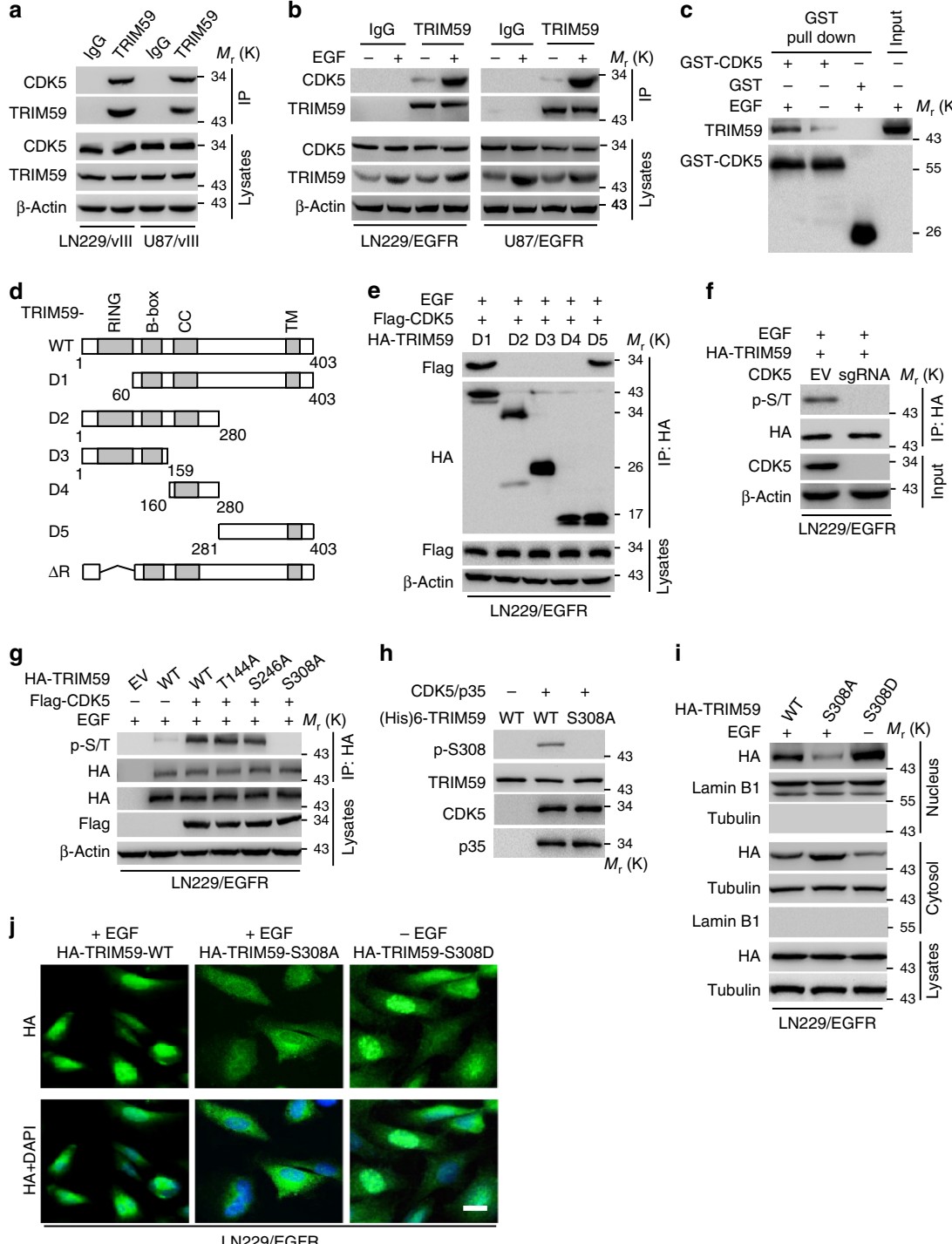

**Fig. 2** CDK5 binds to and phosphorylates TRIM59 at S308. **a** Immunoprecipitation (IP) and WB for CDK5 binding to TRIM59 in LN229/EGFRvIII and U87/EGFRvIII cells. **b** IP and WB for CDK5 binding to TRIM59 in LN229/EGFR and U87/EGFR cells with or without EGF stimulation. **c** In vitro GST pull-down analysis. Purified GST-CDK5 or GST proteins were mixed with cell extracts from U87/EGFR cells with or without EGF stimulation for 30 min. **d** Schematics of TRIM59 WT and truncated constructs. **e** CDK5 interacts with TRIM59 with amino acid residues 281–403. TRIM59 WT or indicated mutants was co-transfected into LN229/EGFR cells with Flag-tagged CDK5 followed by EGF stimulation. **f** Effect of CDK5 knockout on p-Ser/Thr (p-S/T) of TRIM59 in LN229/EGFR GBM cells with EGF stimulation. **g** Effects of mutations of TRIM59 T144A, S246A, and S308A on TRIM59 phosphorylation in LN229/EGFR cells expressing Flag-CDK5 with EGF stimulation. **h** In vitro kinase assays of recombinant active CDK5/p35 with purified TRIM59 WT or S308A mutant as a substrate. The reaction samples were assessed by WB performed with anti-p-TRIM59$^{S308}$, anti-TRIM59, anti-CDK5, or anti-p35 antibody. **i** Localization of TRIM59 WT, S308A, and S308D mutant in LN229/EGFR cells. S308D, a constitutive phosphorylated S308 mutant. **j** IF analysis for TRIM59 WT, S308A, and S308D mutant in LN229/EGFR cells with or without EGF stimulation. Scale bar, 40 μm. Data are representative of three independent experiments with similar results. Source data are provided as a Source Data file

LN229/EGFR cells. Next, we performed in silico analysis for consensus AA residues of TRIM59 as potential serine/threonine kinase substrate using the GPS 3.0 software (http://gps.biocuckoo.org). Three putative CDK5 phosphorylation sites, T144, S246, and S308, were found containing a canonical (Ser/Thr)PX(K/H/R) motif[27] for CDK5 (Supplementary Fig. 5a). To identify the CDK5 phosphorylation site(s) of TRIM59, we co-expressed Flag-tagged CDK5 with or without TRIM59 WT, T144A, S246A, and S308A mutants, or an empty vector in LN229/EGFR cells. Compared with TRIM59 WT, T144A, and S246A mutants, the S308A mutation significantly diminished EGF-stimulated TRIM59 p-Ser/Thr in LN229/EGFR cells (Fig. 2g).

To further evaluate CDK5 phosphorylation of TRIM59 at S308 (p-TRIM59$^{S308}$), we generated a rabbit polyclonal antibody that specifically recognized p-TRIM59$^{S308}$ through a commercial vendor and validated its specificity for p-TRIM59$^{S308}$ in LN229/EGFRvIII GBM cells and a clinical GBM specimen (Supplementary Fig. 5b and 5c). Next, we performed an in vitro kinase assay using purified recombinant active CDK5/p35 and recombinant TRIM59 WT or TRIM59 S308A mutant. CDK5/p35 phosphorylated TRIM59 WT but not the S308A mutant (Fig. 2h), suggesting that the S308 residue is a CDK5 phosphorylation site in TRIM59. Additionally, S308 is found highly conserved in TRIM59 among various species (Supplementary Fig. 5d). These data ares consistent with our result that CDK5 binds to the fragment of TRIM59 AA 281–403 residues that includes residue S308 (Fig. 2e). These data are also in line with the result of a proteomics study showing that S308 of TRIM59 is phosphorylated in ovarian tumors[28].

Lastly, we assessed the effect of phosphorylation of TRIM59$^{S308}$ on TRIM59 nuclear translocation by separately expressing exogenous TRIM59 WT, the non-phosphorylatable S308A, or a phosphor-mimic S308D mutant in LN229/EGFR cells. As shown in Fig. 2i, j, the S308A mutation inhibited the EGF-induced nuclear translocation of TRIM59, whereas TRIM59 WT or the S308D mutant accumulated in the nucleus upon EGF stimulation. Taken together, our data indicate that CDK5 directly binds to and phosphorylates TRIM59 at S308, and p-TRIM59$^{S308}$ is critical for TRIM59 nuclear translocation.

**Phosphorylation of TRIM59 at S308 recruits PIN1.** Proline (Pro)-directed Ser/Thr phosphorylation has crucial roles in protein conformational changes and signal transduction[29]. PIN1 is a unique peptidyl-prolyl cis–trans isomerase that specifically regulates the conformation of p-Ser/Thr-Pro motifs[29,30]. Thus, we sought to determine the potential role of PIN1 in EGFR-promoted TRIM59 nuclear translocation. We found that EGF stimulation markedly enhanced TRIM59 interaction with PIN1 in LN229/EGFR and U87/EGFR GBM cells (Fig. 3a), whereas CDK5 inhibitor Roscovitine attenuated TRIM59–PIN1 association (Fig. 3b). Moreover, in LN229/EGFR cells, EGF stimulation promoted TRIM59 WT, but not S308A, association with PIN1 (Fig. 3c). The phosphor-mimic S308D mutant of TRIM59 was able to associate with PIN1 without EGF stimulation (Fig. 3c). These results suggest that phosphorylation of TRIM59$^{S308}$ is required for TRIM59 association with PIN1 in GBM cells.

Given the PIN1 WW domain binds to p-Ser/Thr-Pro motifs[29], we examined whether the association of TRIM59 and PIN1 depends on PIN1 WW domain. We co-expressed HA-tagged TRIM59 with Flag-PIN1 WT or WW mutant (with substitutions at W11A, W34A, R14A, and R17A)[31] in LN229/EGFR cells and found that PIN1 WW mutant abrogated the association of PIN1 with TRIM59 (Fig. 3d). To further investigate whether the phosphorylated TRIM59 S308/P309 motif is a substrate of PIN1, we synthesized phosphorylated and non-phosphorylatable S308/

P309 peptides of TRIM59. As shown in Fig. 3e, the phosphorylated S308/P309 peptide was more efficiently isomerized by purified WT PIN1 compared with catalytically inactive PIN1 C113A mutant. In contrast, non-phosphorylatable S308/P309 peptide failed to be isomerized by WT PIN1 (Fig. 3e).

Next, we assessed the importance of PIN1 on TRIM59 nuclear translocation. As shown in Fig. 3f, knockout of PIN1 significantly decreased EGF-stimulated nuclear accumulation of TRIM59 in LN229/EGFR cells. Furthermore, we found that nuclear translocation of TRIM59 WT or S308D mutant but not S308A was markedly reduced in LN229/EGFRvIII cells treated with Juglone inhibitor of PIN1 (Supplementary Fig. 6). Additionally, re-expression of PIN1 WT, but not C113A mutant in LN229/EGFR that endogenous PIN1 was knocked down by a short hairpin RNA (shRNA), restored TRIM59 nuclear translocation in PIN1-knockdown cells following EGF stimulation (Fig. 3g).

**PIN1 promotes p-TRIM59$^{S308}$ association with importin α5.** PIN1 contains a nuclear localization signal (NLS) and its R68/R69 residues are critical for the NLS[32]. Thus, we determined whether TRIM59 nuclear translocation is dependent on the NLS of PIN1. As shown in Supplementary Fig. 7, PIN1-knockdown-inhibited EGF-stimulated TRIM59 nuclear translocation in LN229/EGFR cells. Re-expression of PIN1 WT or NLS R68/69A mutant restored PIN1-knockdown-inhibited TRIM59 nuclear translocation, suggesting that nuclear translocation of TRIM59 is not dependent on the NLS of PIN1.

Next, to explore whether TRIM59 itself harbors an NLS and its NLS is exposed through PIN1-mediated cis–trans isomerization, we performed in silico analysis using the NLStradamus software (http://www.moseslab.csb.utoronto.ca/NLStradamus/)[33] and identified a putative NLS sequence in the carboxy terminus, 301-LIPKMKISPKRMS-313, and the sequence is conserved in various species (Fig. 4a). We then expressed exogenous TRIM59 WT or a NLS mutant in LN229/EGFR cells (Fig. 4b). As shown in Fig. 4b, c, ectopic expression of the TRIM59 NLS mutant, but not WT, reduced EGF-stimulated nuclear translocation of TRIM59. These results indicate that the NLS of TRIM59 is critical for EGF-induced nuclear translocation of TRIM59.

Nuclear transport of NLS-containing protein is mediated by importin α/β1 transport system, in which importin α protein recognizes and links NLS sequences[29]. Thus, we separately co-expressed HA-TRIM59 with importin α1, α3, α4, α5, α6, or α7 in LN229/EGFR cells and found that with EGF stimulation, TRIM59 specifically bound to importin α5 but not other importins tested (Fig. 4d). Furthermore, shRNA knockdown of importin α5 abrogated EGF-induced TRIM59 nuclear translocation (Fig. 4e). Additionally, when the Flag-tagged importin α5 was co-expressed with TRIM59 WT, S308A, or NLS mutant in LN229/EGFR cells, TRIM59 S308A and NLS mutants, but not WT, reduced EGF-stimulated TRIM59 interaction with importin α5 (Fig. 4f). These data suggest that PIN1-promoted p-TRIM59$^{S308}$ nuclear translocation is dependent on importin α5.

Finally, we examined whether PIN1 is essential for the binding of TRIM59 to importin α5. We separately co-expressed Myc-importin α5 and a phosphor-mimic HA-TRIM59 S308D mutant with Flag-PIN1 WT or PIN1 C113A mutant in LN229 cells. We found that PIN1 WT increased TRIM59 S308D association with importin α5, whereas PIN1 C113A mutant attenuated their interaction (Fig. 4g). These results indicate that PIN1-mediated cis–trans isomerization of p-TRIM59$^{S308}$ exposes the TRIM59 NLS sequence for TRIM59 binding to importin α5, suggesting that the PIN1/importin α5 axis is required for TRIM59 nuclear translocation.

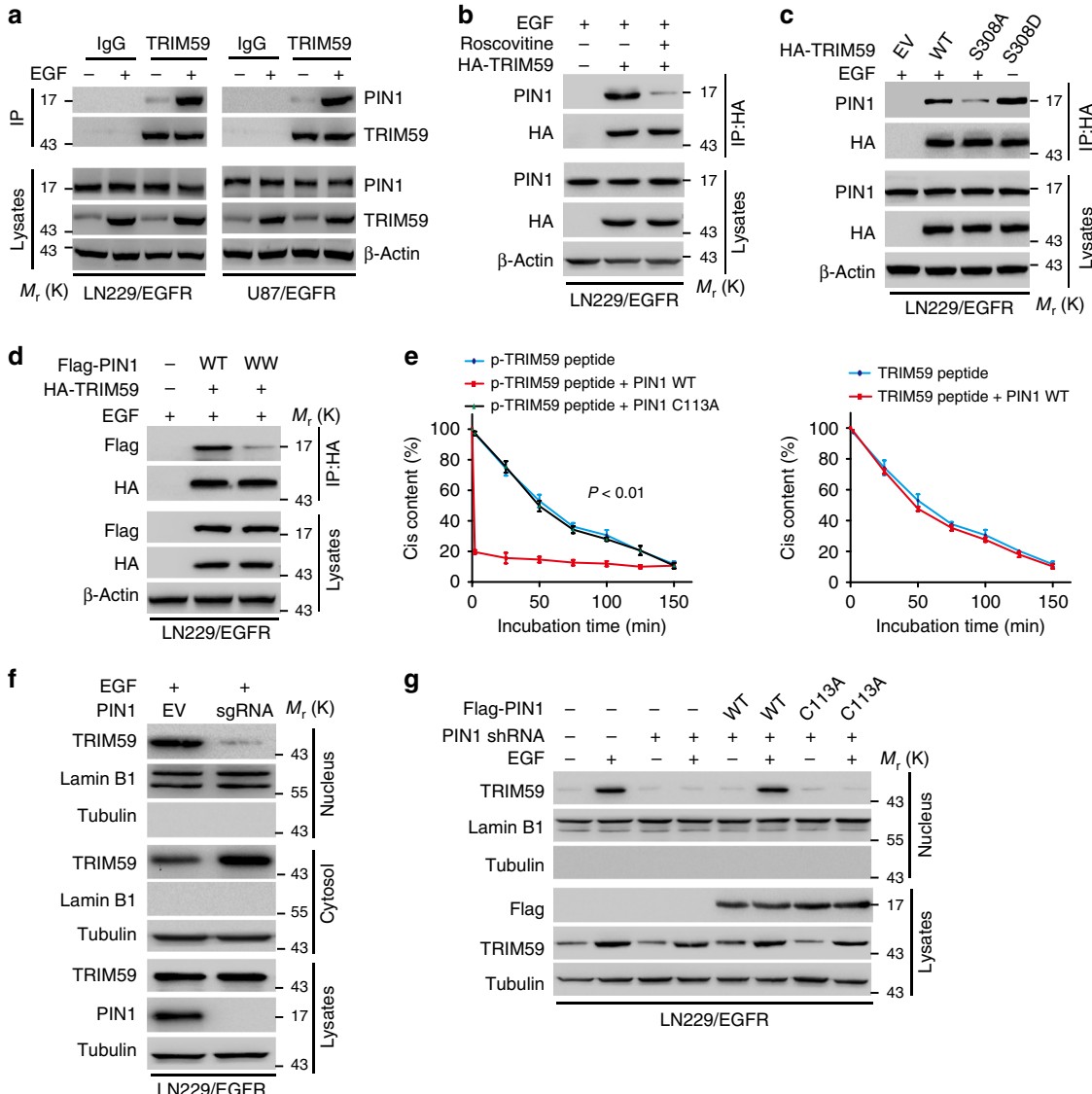

**Fig. 3** Phosphorylation of TRIM59 at Ser308 recruits PIN1. **a** IP and WB for TRIM59 binding to PIN1 in LN229/EGFR and U87/EGFR cells with or without EGF stimulation. **b** IP and WB for TRIM59 association with PIN1 in LN229/EGFR cells stimulated with EGF (100 ng/ml) for 6 h. Cells were pre-treated with or without Roscovitine (20 μM) for 1 h. **c** Effects of ectopic expression of TRIM59 WT, S308A, or S308D mutant on TRIM59 association with PIN1 with or without EGF stimulation. **d** Effects of ectopic expression of PIN1 WT or WW mutant on TRIM59 binding with PIN1. **e** Cis–trans isomerization analysis. The synthesized phosphorylated or nonphosphorylated oligopeptides of TRIM59 were mixed with purified PIN1 WT or C113A mutant. Data were expressed as means ± SD. P values were calculated using one-way analysis of variance (ANOVA). **f** WB for effect of PIN1 knockout with a PIN1 sgRNA on TRIM59 nuclear translocation in LN229/EGFR GBM cells. **g** Effects of re-expression of shRNA-resistant PIN1 WT or C113A mutant on TRIM59 nuclear translocation. LN229/EGFR GBM cells reconstituted with empty vector (EV), shRNA-resistant PIN1 WT, or C113A mutant were stimulated with or without EGF (100 ng/ml) for 6 h. Data are representative of three independent experiments with similar results. Source data are provided as a Source Data file

**Nuclear TRIM59 associates with macroH2A1**. To gain insights into the mechanism by which nuclear TRIM59 promotes GBM tumorigenicity, we purified the TRIM59 complex using HA pull-down from the nuclear extract of EGF-treated LN229/EGFR cells transduced with HA-tagged TRIM59 or an empty vector control, followed by mass spectrometry. Among a total of 70 putative TRIM59-binding proteins identified in our analysis, we found a total of seven macroH2A1 (mH2A1) peptides in the protein complex (Fig. 5a and Supplementary Fig. 8). Therefore, mH2A1 was selected for the further studies. mH2A1 is generally enriched on condensed chromatin such as the inactive X chromosome[34,35] and represses gene transcription[36,37]. Moreover, recent evidence suggests that mH2A1, consisting of two isoforms mH2A1.1 and

mH2A1.2, is a potential tumor suppressor in various types of human cancers[38].

Thus, we validated TRIM59 interaction with mH2A1 by IP assays. Our data revealed that EGF stimulated the association of endogenous TRIM59 with mH2A1.1 and mH2A1.2 in LN229/EGFR and U87/EGFR GBM cells (Fig. 5b). Our data of IF assays further demonstrated that EGF stimulated p-TRIM59[S308] co-localization with mH2A1 in the nucleus in U87/EGFR GBM cells (Fig. 5c). Additionally, re-expression of shRNA-resistant TRIM59 S308A or NLS mutant impaired the association of TRIM59 with mH2A1 (Fig. 5d). Interestingly, compared with the empty vector control, re-expression of TRIM59 WT decreased mH2A1 protein levels in U87/EGFR/shTRIM59 cells (Fig. 5d), suggesting that

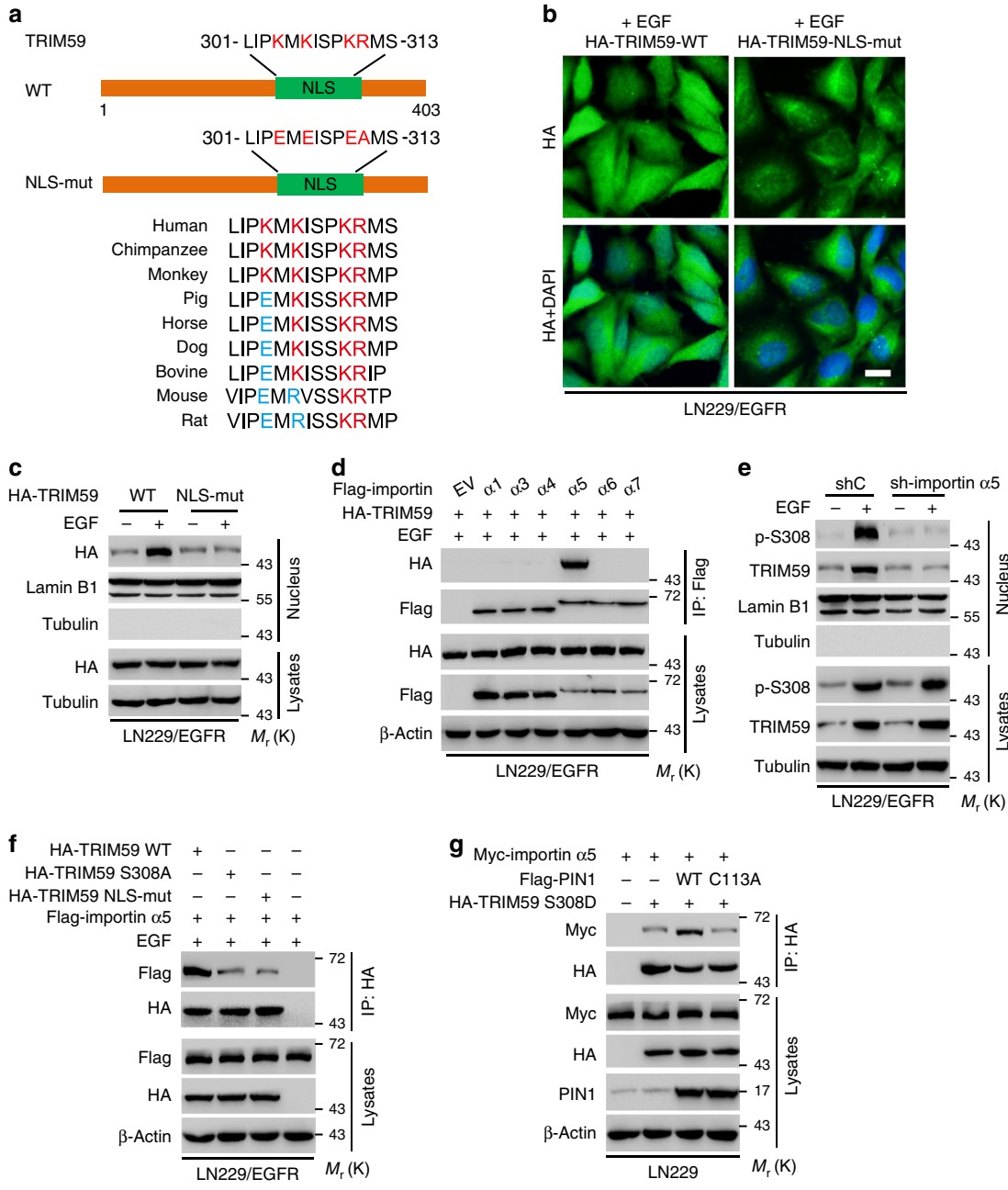

**Fig. 4** PIN1 promotes p-TRIM59$^{S308}$ association with importin α5. **a** Schematics of TRIM59 WT and nuclear localization sequence (NLS) mutant (NLS-mut). **b** Representative images of TRIM59 nuclear translocation in LN229/EGFR cells resconstituted with TRIM59 WT or NLS mutant. Scale bar, 40 μm. **c** WB for effect of TRIM59 WT or NLS-mut on TRIM59 nuclear translocation. **d** IP and WB for TRIM59 binding to importin α proteins indicated. Flag-tagged importin α proteins and HA-tagged TRIM59 were co-expressed in LN229/EGFR cells. **e** Effect of importin α5 knockdown on TRIM59 nuclear translocation. **f** Effects of TRIM59 WT, S308A, or NLS-mut expression on TRIM59 binding with importin α5. Flag-tagged importin α5 was transfected into LN229/EGFR cells with HA-TRIM59 WT, S308A, NLS-mut mutant, or an empty vector control. **g** WB for effects of ectopic expression of PIN1 WT or C113A mutant on TRIM59 binding with importin α5. In **b–f**, various cells were stimulated with or without EGF (100 ng/ml) for 6 h. Data are representative of three independent experiments with similar results. Source data are provided as a Source Data file

TRIM59 may regulate mH2A1 expression in a transcriptional or post-transcriptional level.

**Nuclear TRIM59 promotes mH2A1 ubiquitination and degradation.** To further investigate the mechanism by which nuclear TRIM59 regulates mH2A1 expression, we assessed the effects of TRIM59 knockdown on the expression of mH2A1 protein and mRNA in LN229/EGFR and U87/EGFR cells with

EGF stimulation. As shown in Fig. 6a, b, TRIM59 knockdown increased levels of mH2A1 protein without affecting the levels of mH2A1 mRNAs. Given the critical role of PIN1 and importin α5 in TRIM59 nuclear translocation, we separately knocked down PIN1 or importin α5. As expected, levels of mH2A1 protein were increased after knockdown of PIN1 or importin α5 (Supplementary Fig. 9a, b). On the other hand, overexpression of TRIM59 decreased the levels of mH2A1 protein but not its mRNAs (Supplementary Fig. 10a, b). In combination with the

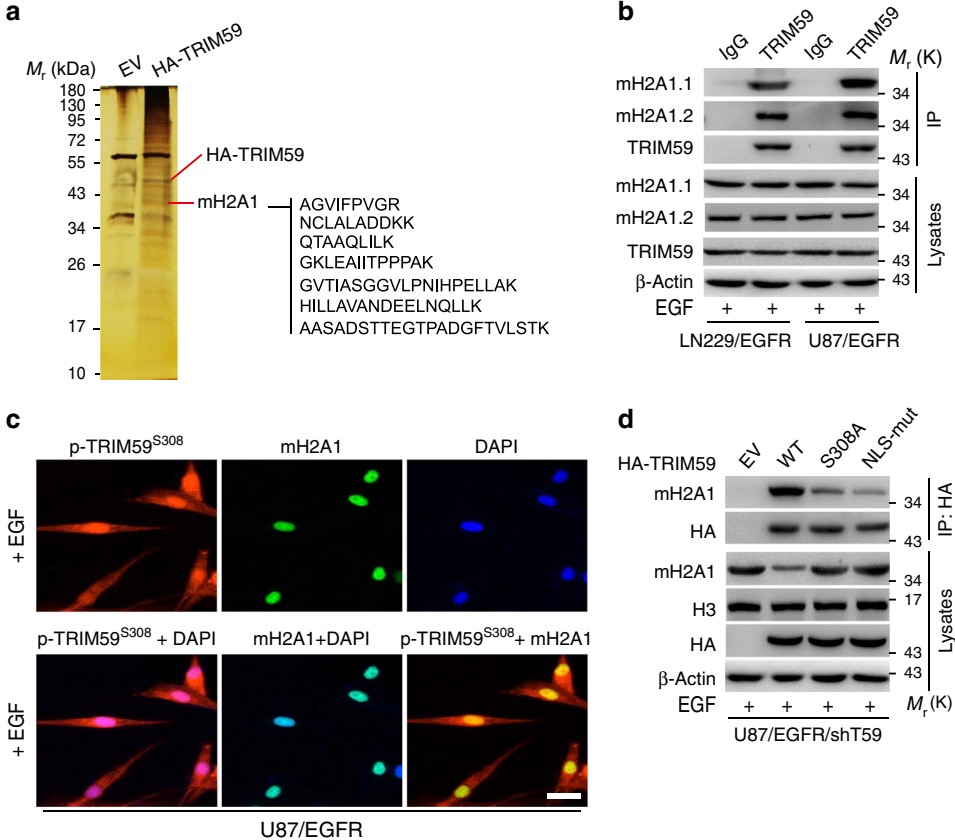

**Fig. 5** Nuclear TRIM59 associates with macroH2A1. **a** Silver staining showing the nuclear TRIM59-associated proteins in LN229/EGFR cells with EGF stimulation. **b** IP and WB for TRIM59 binding with macroH2A1.1 (mH2A1.1) and macroH2A1.2 (mH2A1.2) proteins. **c** Representative images of co-localization of TRIM59 with mH2A1. Scale bar, 40 μm. **d** WB for effects of TRIM59 S308A and NLS-mut on TRIM59 binding with mH2A1 in U87/EGFR/shTRIM59 cells. In **b**–**d**, various cells were stimulated with or without EGF (100 ng/ml) for 6 h. Data are representative of three independent experiments. Source data are provided as a Source Data file

data shown in Fig. 5d, our results suggest that TRIM59 regulates the abundance of mH2A1 protein through post-transcriptional mechanisms.

Since TRIM59 is a ubiquitination ligase[19,20], we determined whether TRIM59 ubiquitinates mH2A1, resulting in mH2A1 protein degradation. Consistent with this prediction, knockdown of TRIM59 significantly attenuated the degradation of endogenous mH2A1 in LN229/EGFR (Fig. 6c, d) and U87/EGFR cells (Fig. 6e, f) with EGF stimulation, whereas TRIM59 overexpression markedly accelerated the degradation of endogenous mH2A1 (Supplementary Fig. 10c–f). Furthermore, compared with the controls, exogenous TRIM59 WT and S308D, but not S308A, markedly increased the ubiquitination of mH2A1.1 (Fig. 6g) and mH2A1.2 (Supplementary Fig. 11) in LN229/EGFR and U87/EGFR cells with EGF stimulation. Furthermore, CDK5 inhibitor Roscovitine treatment decreased mH2A1 ubiquitination regulated by TRIM59 WT but not S308D mutant (Fig. 6h).

The RING domain of TRIM family proteins has been shown to be critical for their E3 ligase activity[39,40]. Thus, we performed in vitro ubiquitination assay and found that TRIM59 WT, but not RING-domain deletion mutant (ΔR), could induce mH2A1 ubiquitination (Fig. 6i). To identify the TRIM59 ubiquitination sites of mH2A1, we first analyzed the published large-scale quantitative proteomics data[41,42] and found that multiple lysine (K) residues (K18, K116, K123, K167, K189, K235, K251, K285, K292, K295) of mH2A1 were ubiquitinated. We further mutated each lysine (K) to arginine (R) in mH2A1 and found that only mH2A1 K167R mutant, but not other KR

mutants, markedly reduced mH2A1 ubiquitination regulated by TRIM59 (Supplementary Fig. 12). These data show that TRIM59 ubiquitinates mH2A1 proteins and reduces their stability.

**TRIM59-mediated mH2A1 degradation enhances STAT3 signaling.** To investigate the role of mH2A1 in GBM tumorigenicity, we performed genome-wide mapping of mH2A1 binding using chromatin immunoprecipitation-sequencing (ChIP-Seq) in LN229/EGFRvIII cells treated with or without EGFR inhibitor erlotinib or CDK5 inhibitor Roscovitine (Fig. 7a). Our data showed that compared with the vehicle treatment cells, mH2A1 bindings in the genomic DNA showed an increase in LN229/EGFRvIII cells treated with erlotinib or Roscovitine (Fig. 7b). We then performed Gene Ontology analysis and found that both treatments of erlotinib and Roscovitine markedly influenced mH2A1 binding at genes from many pathways (Supplementary Table 1). The top five EGFR-driven pathways, including JAK-STAT signaling pathway, were shown (Fig. 7c). Integrative Genomics Viewer (IGV) snapshots of ChIP-Seq tracks further revealed significant increases in mH2A1 binding at the STAT3-targeted genes, *PIM1*[43], *MUC1*[44,45], and *AKT3*[46] following erlotinib or Roscovitine treatment (Fig. 7d). These data were finally validated by ChIP-qPCR (Fig. 7e). Together, these data strongly support our observation that mH2A1 binding is mediated by EGFR and CDK5 activity, and mH2A1 may regulate STAT3 signaling.

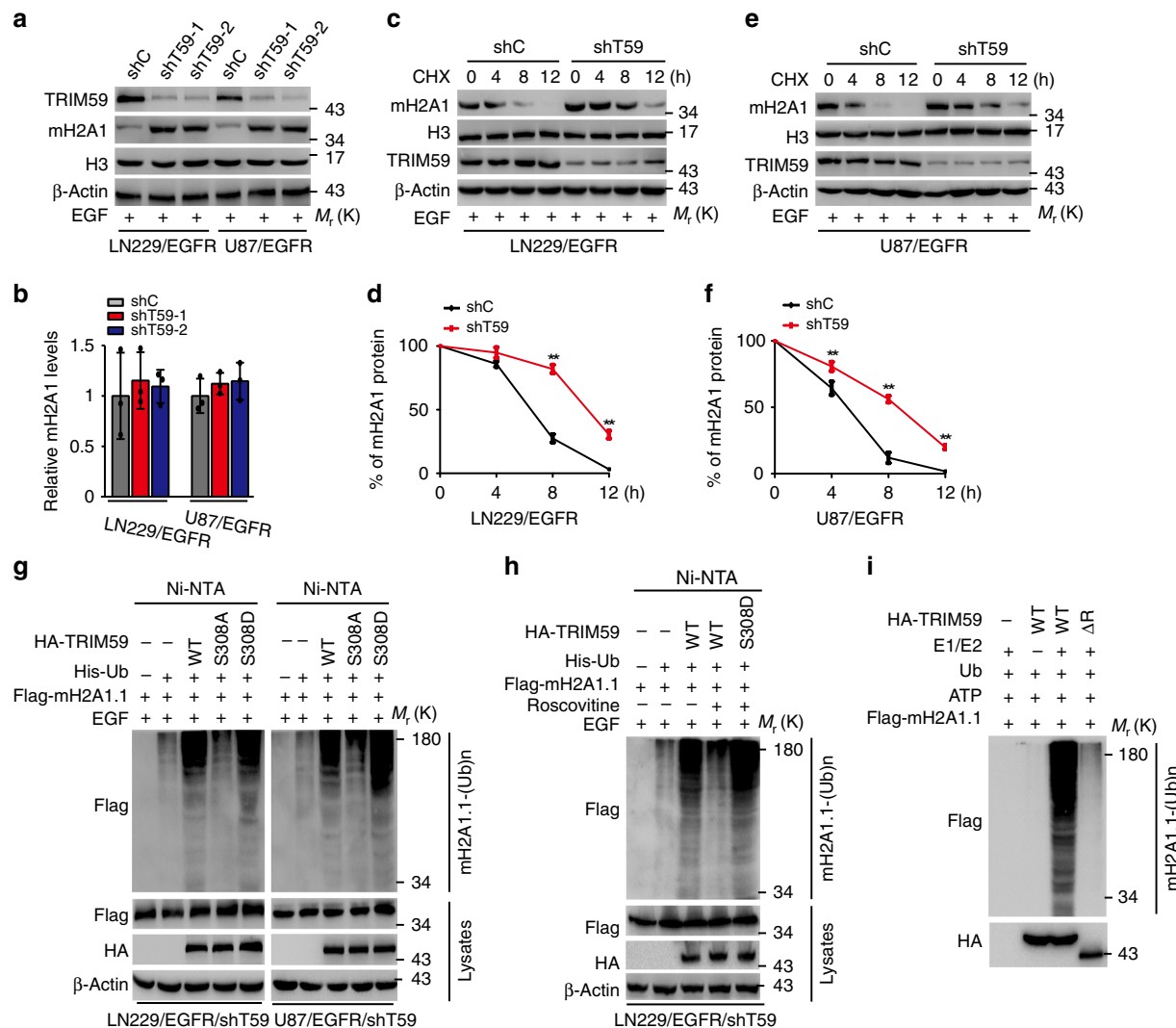

**Fig. 6** Nuclear TRIM59 promotes macroH2A1 ubiquitination and degradation. **a**, **b** WB and qRT-PCR assays of effects of TRIM59 knockdown on macroH2A1 protein (**a**) and mRNA (**b**) levels in LN229/EGFR and U87/EGFR cells stimulated with EGF. **c**, **e** Effects of TRIM59 knockdown on macroH2A1 (mH2A1) stability in LN229/EGFR (**c**) and U87/EGFR (**e**) cells. After stimulation with EGF (100 ng/ml) for 6 h, cells were treated with cycloheximide (CHX, 20 mg/ml) for the indicated time points. **d**, **f** Quantification of mH2A1 protein levels in **c** and **e**, respectively. **g** Effects of TRIM59 WT, S308A, or S308D on mH2A1.1 ubiquitination. His-tagged ubiquitin (His-Ub) was co-transfected into LN229/EGFR/shTRIM59 and U87/EGFR/shTRIM59 cells with TRIM59 constructs or empty vector control. **h** Effects of Roscovitine on mH2A1.1 ubiquitination. His-Ub was co-transfected into LN229/EGFR/shTRIM59 and U87/EGFR/shTRIM59 cells with TRIM59 constructs or empty vector control. **i** HA-TRIM59 proteins purified from HEK293 cells were incubated with ATP, E1, and E2 along with Flag-mH2A1.1 proteins isolated from HEK293 cells for in vitro ubiquitination assay. Data are representative of three independent experiments with similar results. Data were expressed as means ± SD. **$P < 0.01$ by two-tailed Student's $t$ test. Source data are provided as a Source Data file

We recently reported that TRIM59 enhances nuclear STAT3 signaling activity by inhibiting TC45-mediated STAT3 dephosphorylation and Y218/Q221 sites of TRIM59 are critical for TRIM59–STAT3 interaction[21]. Thus, we further determined whether nuclear TRIM59-mediated macroH2A ubiquitination affects nuclear STAT3 signaling activation. Knockdown of mH2A1 did not affect EGF-stimulated TRIM59, STAT3 protein levels, and STAT3 phosphorylation (Fig. 7f) in LN229/EGFR and U87/EGFR cells, but increased STAT3 binding with *PIM1* promoter (Fig. 7g) and *PIM1* mRNA levels (Fig. 7h), suggesting that TRIM59 increases macroH2A1 ubiquitination and degradation, thereby forming a more accessible chromosome structure to facilitate STAT3 binding to its target gene promoter, resulting in enhanced nuclear STAT3 signaling.

Next, to further demonstrate the mechanism by which TRIM59 regulates STAT3 signaling by promoting macroH2A1

ubiquitination and degradation and inhibiting TC45 dephosphorylation in GBM, we compared the effects of exogenous TRIM59 WT, S308A, ΔR, or Y218F mutant on mH2A1 protein levels, p-STAT3, STAT3 binding with the *PIM1* promoter, and *PIM1* mRNA expression in LN229/EGFR/shTRIM59 GBM cells. Compared with the empty vector control, re-expression of TRIM59 WT, but not the non-phosphorylatable S308A mutant, decreased mH2A1 protein levels and increased p-STAT3 (Fig. 7i) after EGF stimulation, leading to enhanced STAT3 binding with the *PIM1* promoter (Fig. 7j) and *PIM1* mRNA expression (Fig. 7k). Re-expression of TRIM59 RING-domain deletion mutant (ΔR) that inhibited TRIM59 E3 ligase activity did not reduce mH2A1 protein levels but increased p-STAT3 (Fig. 7i), whereas it did not enhance STAT3 binding with the *PIM1* promoter (Fig. 7j) and *PIM1* mRNA expression (Fig. 7k). Re-expression of TRIM59 nuclear STAT3 binding site mutant (Y218F) that was not able to

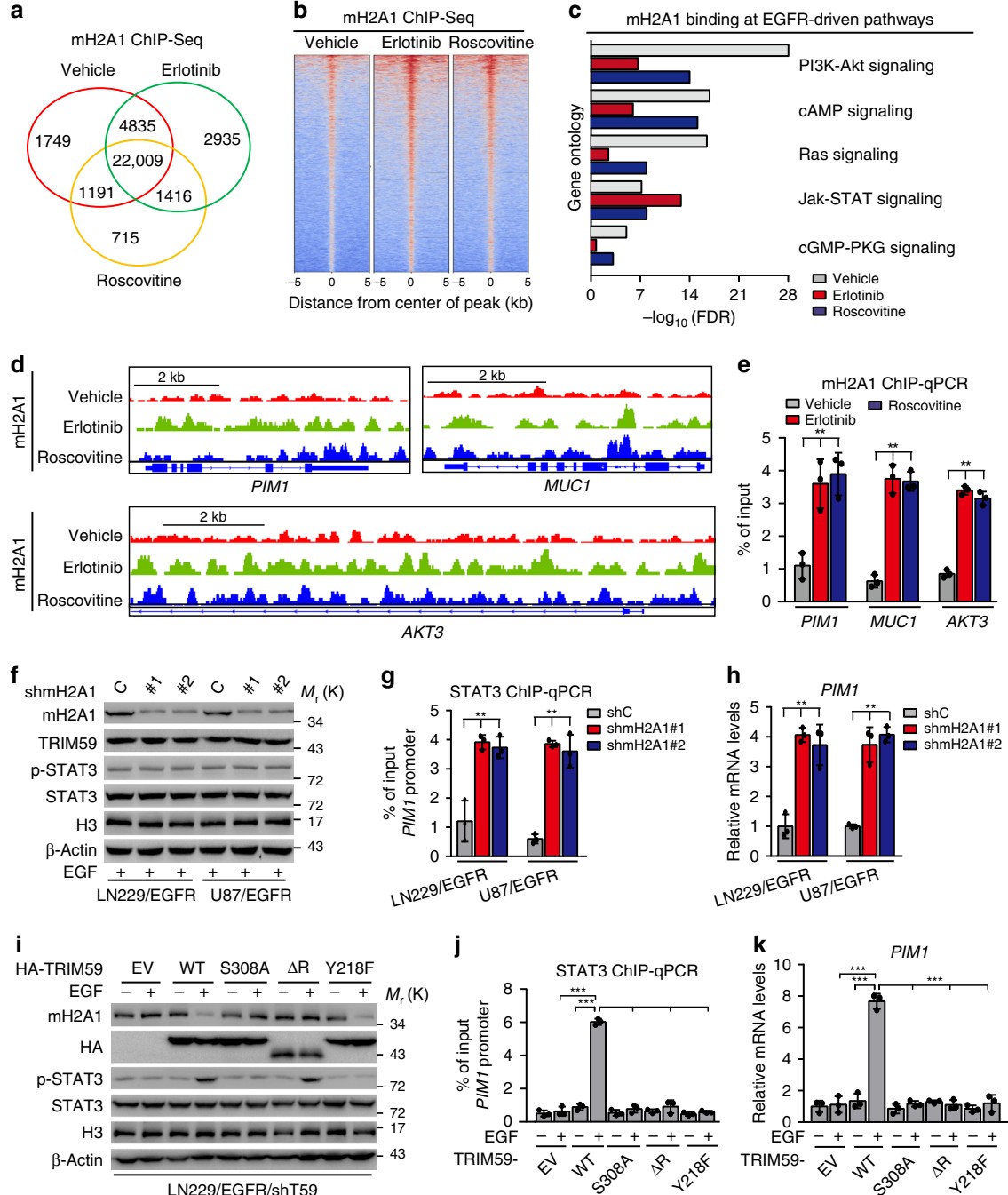

**Fig. 7** TRIM59-mediated macroH2A1 degradation enhances STAT3 signaling. **a** Venn diagram shows the overlap of mH2A1-bound sites in the genome in LN229/EGFRvIII cells treated with vehicle (0.1% DMSO), erlotinib (10 μM), or Roscovitine (20 μM) for 16 h using ChIP-Seq. **b** Heatmap of mH2A1 ChIP-Seq. **c** Gene ontology analysis of effects of both the treatment of erlotinib and Roscovitine on mH2A1 binding in distance from center of peak (±2 kb) at top five EGFR-driven pathways. **d** mH2A1 ChIP-Seq peaks at STAT3-targeted genes using IGV (Integrative Genomics Viewer). **e** ChIP-qPCR of effects of erlotinib or Roscovitine on mH2A1 binding at STAT3-targeted genes. **f–h** Effects of mH2A1 knockdown in LN229/EGFR and U87/EGFR cells with EGF stimulation. WB for expression of TRIM59, STAT3, and p-STAT3 (**f**); ChIP-qPCR analysis for STAT3 binding with *PIM1* promoter (**g**); qRT-PCR analysis for STAT3-targeted gene *PIM1* transcription (**h**). **i–k** Effects of reconstituted TRIM59 WT, S308A, ΔR, or Y218F with or without EGF stimulation. WB for expression of TRIM59, mH2A1, and p-STAT3 (**i**); ChIP-qPCR analysis for STAT3 binding with *PIM1* promoter (**j**); qRT-PCR analysis for *PIM1* transcription (**k**). Data are representative of three independent experiments with similar results. Data were expressed as means ± SD. **P < 0.01, ***P < 0.001, by two-tailed Student's t test. Source data are provided as a Source Data file

inhibit TC45 dephosphorylation of STAT3[21] decreased mH2A1 protein levels but not affect p-STAT3 (Fig. 7i), which also did not enhance STAT3 binding with the *PIM1* promoter (Fig. 7j) and *PIM1* mRNA expression (Fig. 7k).

Additionally, consistent with our previous report[21], re-expression of TRIM59 WT diminished TC45 association with STAT3 and increased p-STAT3 compared with the empty vector control in U87/EGFRvIII/shTRIM59 GBM cells (Supplementary

Fig. 13). However, re-expression of non-phosphorylatable TRIM59 S308A or NLS mutant that inhibited TRIM59 nuclear translocation did not affect TC45 association with nuclear STAT3 and p-STAT3, further suggesting that TRIM59 inhibition of TC45 dephosphorylation of STAT3 depends on TRIM59 nuclear translocation (Supplementary Fig. 13). Taken together, our data indicate that nuclear TRIM59-mediated STAT3 signaling activation depends on not only inhibiting TC45 dephosphorylation of nuclear STAT3 to sustain high p-STAT3 but also promoting mH2A1 ubiquitination and degradation to facilitate STAT3 binding to downstream target gene promoters.

**CDK5/TRIM59 axis is required for GSC tumorigenicity.** To further demonstrate whether CDK5-promoted TRIM59 nuclear translocation is critical for glioma tumorigenicity, we analyzed the effects of EGFR inhibitor erlotinib and CDK5 inhibitor Roscovitine treatments on patient-derived glioma stem-like cells (GSCs) and primary GBM cells using established methods to evaluate cell signaling pathways, self-renewal, and tumor-forming ability[47,48]. GSC R39 cells express high levels of endogenous EGFRvIII; GSC R83, GBM6, and GBM15 cells express high levels of endogenous EGFR WT (Fig. 8a and Supplementary Fig. 14a). Treatment of erlotinib and Roscovitine significantly inhibited EGFR phosphorylation (p-EGFR) and/or p-CDK5 in GSC R39, R83, GBM6, and GBM15 cells, respectively (Fig. 8a and Supplementary Fig. 14a). Both erlotinib and Roscovitine treatments markedly inhibited TRIM59 nuclear translocation (Fig. 8a and Supplementary Fig. 14a), p-STAT3 (Fig. 8a and Supplementary Fig. 14a), PIM1 mRNA expression (Fig. 8b and Supplementary Fig. 14b), and glioma sphere formation (Fig. 8c), but increased mH2A1 protein levels (Fig. 8a and Supplementary Fig. 14a), validating our observations in LN229/EGFR GBM cells (Fig. 1a, d).

To further validate the signaling pathway of CDK5/TRIM59/mH2A1/STAT3, we re-expressed different shRNA-resistant TRIM59 constructs, WT, the non-phosphorylatable S308A, or the phosphor-mimic S308D mutant in GSC R39, GSC R83, GBM6, and GBM15 cells with a TRIM59 shRNA. Compared with the shRNA control, knockdown of TRIM59 significantly inhibited p-STAT3 (Fig. 8d and Supplementary Fig. 14c), PIM1 mRNA expression (Fig. 8e and Supplementary Fig. 14d), glioma sphere formation (Fig. 8f), and tumorigenicity of intracranial xenografts (Fig. 8g, h), but increased mH2A1 protein levels (Fig. 8d and Supplementary Fig. 14c). Re-expression of shRNA-resistant TRIM59 WT or S308D mutant rescued p-STAT3 (Fig. 8d and Supplementary Fig. 14c), PIM1 mRNA expression (Fig. 8e and Supplementary Fig. 14d), glioma sphere formation (Fig. 8f), and tumorigenesis of intracranial xenografts (Fig. 8g, h), but inhibited mH2A1 protein levels (Fig. 8d and Supplementary Fig. 14c). However, re-expression of TRIM59 S308A mutant was unable to restore these effects (Fig. 8d–h). Compared with the shRNA control, TRIM59 knockdown decreased expression of p-STAT3, proliferation marker Ki-67, and cancer cell stemness marker CD44, whereas the levels of mH2A1 was elevated in GSC xenograft tumor tissues by immunohistochemical (IHC) analyses. Re-expression of shRNA-resistant TRIM59 WT or S308D but not S308A mutant in TRIM59-knockdown cells rescued levels of p-STAT3, Ki-67, and CD44 expression while reducing mH2A1 levels (Fig. 8i). These were further validated by WB and quantitative reverse transcription-PCR (qRT-PCR) (Fig. 8j, k). Together, these data indicate that EGFR-stimulated CDK5-dependent phosphorylation and nuclear translocation of TRIM59 enhances STAT3 signaling activation and tumorigenicity by regulating macroH2A1 protein levels.

**CDK5/TRIM59/STAT3 pathway is clinically prognostic.** To further investigate the clinical relevance of our findings, we examined the expression of p-EGFR, p-CDK5, p-TRIM59, and p-STAT3 by IHC analyses in a cohort of 120 clinical GBM specimens. Co-expression of p-CDK5, p-TRIM59, and p-STAT3 was found in the majority of p-EGFR-expressing tumors (Fig. 9a). Quantification of the IHC staining indicated that these correlations were statistically significant (Fig. 9b and Supplementary Fig. 15a). Moreover, co-expression of p-CDK5/p-TRIM59, p-CDK5/p-STAT3, or p-TRIM59/p-STAT3 at high levels correlated with GBM prognosis (Fig. 9c). High-level p-CDK5 or p-TRIM59 in p-EGFR-expressing GBM patients was found to be associated with poor outcome (Supplementary Fig. 15b).

**Discussion**
Our data describe a CDK5–TRIM59 signaling axis that enhanced STAT3 signaling activation and EGFR-driven tumorigenicity not only through maintaining nuclear STAT3 phosphorylation by inhibiting TC45 dephosphorylation but also through promoting macroH2A1 ubiquitination and degradation (Fig. 9d). The prevalence of EGFR as an oncogenic driver across many prominent types of cancers including GBM renders EGFR an appealing target for therapeutic intervention[49]. However, a number of mechanisms have been proposed to underlie GBM resistance to EGFR-targeted therapies. One such mechanism is that EGFR inhibition induces a rapid adaptive response that mediates resistance to EGFR inhibition[50]. On the other hand, it has been recognized that EGFR signaling network in GBM is highly heterogeneous[2]. Our findings in this study provide novel insights of the mechanisms underlying EGFR-driven GBM tumorigenic phenotype. This new knowledge will be highly useful for developing effective treatments for EGFR expressing GBM using a combination inhibition of EGFR and the newly described CDK5/TRIM59 axis.

This study demonstrates that CDK5 promotes GBM tumor growth through TRIM59-mediated STAT3 signaling activation. CDK5 is known to be critical for tumorigenesis and cancer progression[7–9]. Genetic approaches had shown that Drosophila Cdk5 is overexpressed in Drosophila brain tumor stem cells and regulates asymmetric cell division of neuroblasts as well as tumor growth[16]. In clinical specimens, increased levels of CDK5 in GBM associated with poor patient prognosis[16,18]. Moreover, pharmacological inhibition of CDK5 activity attenuated CDK5 promotion of GSC self-renewal in vitro[16]. This study not only corroborates these studies but further advances our knowledge of the role of CDK5 in cancer. We showed that CDK5 regulates glioma tumorigenicity by promoting TRIM59 nuclear translocation via PIN1/importin α5 axis, leading to STAT3 signaling activation, and demonstrate that CDK5 mediates EGFR-stimulated STAT3 signaling through activation of the TRIM59/mH2A1 axis in GBM.

This study identified and validated TRIM59 as a novel substrate of CDK5. Accumulated data have shown that CDK5 is an atypical CDK and critical for tumor growth in several types of human cancers[9,11]. Thus far, however, only a few proteins including PIKE-A, FAK, and DRP1 have been identified as CDK5 substrates[14,18,51]. TRIM59 is a member of the TRIM) protein superfamily that functions as a ubiquitin ligase of p53 in gastric tumors[19] or TRAF6 in non-small-cell lung cancer[20] or as an adaptor protein in the Toll-like receptor-mediated transduction pathway in HeLa cells[22]. TRIM59 is critical for tumor growth in human prostate cancer, lung cancer, osteosarcoma, and cervical cancer[23,24,52]. We recently reported that TRIM59 promoted GBM tumorigenicity by inhibiting TC45 dephosphorylation of nuclear STAT3[21]. In this study, we further identified TRIM59 as

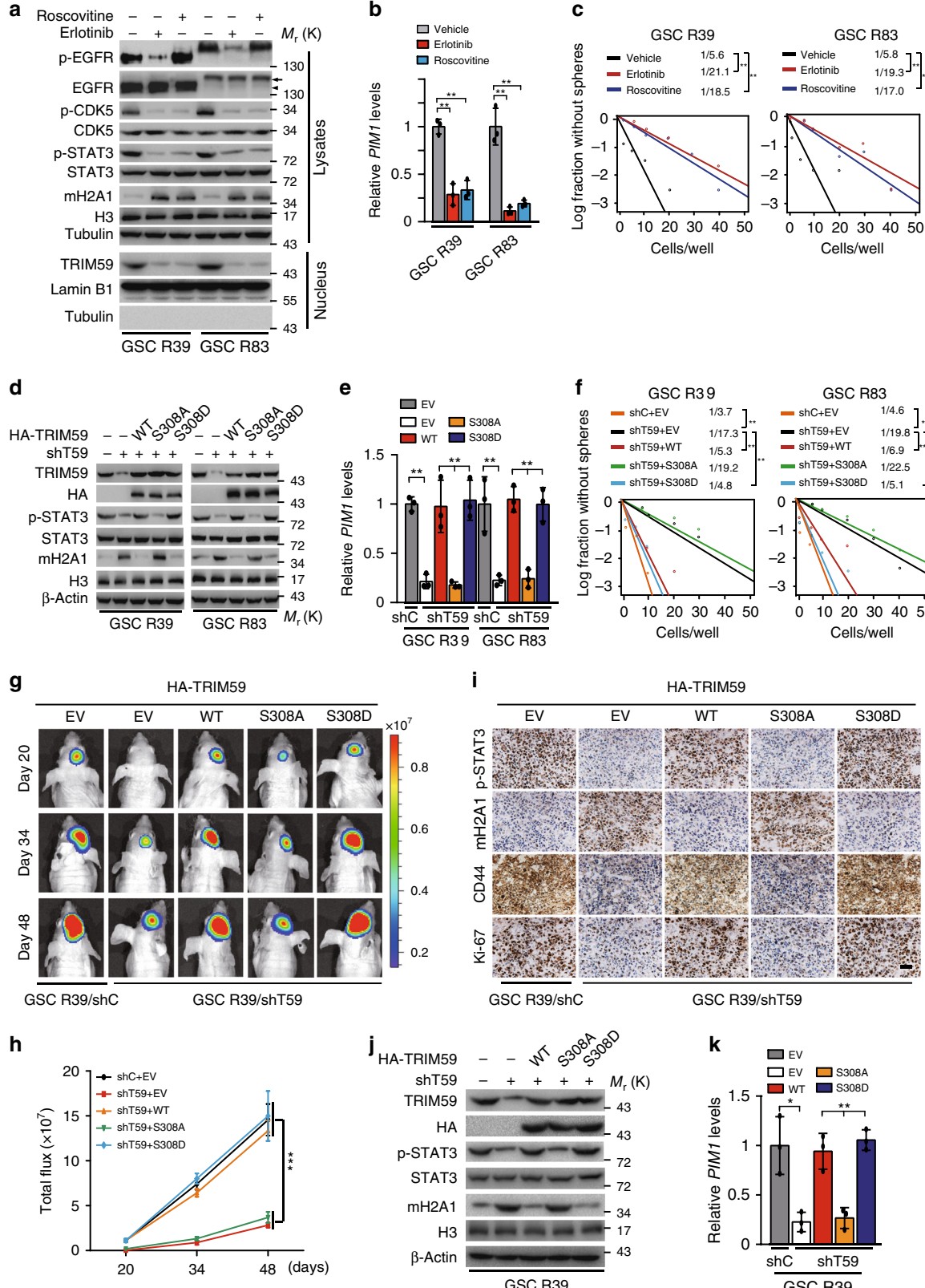

an authentic substrate of CDK5. CDK5 phosphorylation of TRIM59 at S308 promotes TRIM59 nuclear translocation, STAT3 signaling, GSC self-renewal, and GBM tumorigenicty. Moreover, p-TRIM59[S308] has a prognostic value in combination with the expression of p-CDK5 or p-STAT3 in predicting poor clinical outcomes of GBM.

This study also demonstrated that TRIM59 mediates mH2A1 ubiquitination and degradation and thereby activates STAT3 signaling. As histone H2A variants, mH2A1 proteins are structural chromatin components and mediate gene expression[53]. Although mH2A1 proteins promote gene expression[54], accumu- lated data revealed that macroH2A1 proteins regulate gene

**Fig. 8** TRIM59 nuclear translocation is required for GSC tumorigenicity. **a** Effects of EGFR inhibitor erlotinib or CDK5 inhibitor Roscovitine treatment on p-EGFR, p-CDK5, p-STAT3, and mH2A1 protein levels in GSC R39 and R83 cells. Arrows, EGFR. Arrow heads, EGFRvIII. **b** qRT-PCR analyses of erlotinib or Roscovitine treatment on STAT3-targeted gene *PIM1* mRNA expression. **c** Limiting dilution glioma sphere-forming assays in GSC R39 and R83 cells with erlotinib or Roscovitine treatment. **d**–**f** Effects of TRIM59 shRNA knockdown and re-expression of shRNA-resistant TRIM59 WT, S308A, or S308D mutant on TRIM59, mH2A1 protein levels (**d**), p-STAT3 (**d**), STAT3-targeted gene *PIM1* expression (**e**), and glioma sphere formation (**f**) in GSC R39 and R83 cells. **g**, **h** Representative bioluminescence images of tumor growth (**g**) and quantification of bioluminescence (**h**). *n* = 5. **i** Representative images of IHC analysis of indicated protein expression in GSC R39 xenograft tumors. Scale bar, 50 μm. **j**, **k** Effects of TRIM59 shRNA knockdown and re-expression of shRNA-resistant TRIM59 WT, S308A, or S308D mutant on TRIM59, mH2A1 protein levels (**j**), p-STAT3 (**j**), and STAT3-targeted gene *PIM1* expression (**k**) in GSC R39 xenograft tumors. Data are representative of three independent experiments with similar results. Data were expressed as means ± SD. *P* < 0.05, **P* < 0.01, by two-tailed Student's *t* test. Source data are provided as a Source Data file

repression by interfering with the binding of transcription and nucleosome remodeling factors[54–56]. Thus, regulation of mH2A1 stability alters the chromatin structure and gene expression. However, only several proteins including SKP2, BRCA1, and CULLIN3/SPOP were identified as an E3 ligase of mH2A1 proteins[57–59]. In this study, we described that nuclear TRIM59 associated with mH2A1 and regulated mH2A1 proteins ubiquitination and stability as an E3 ligase in GBM. A large number of mH2A1-bound chromatin regions were identified using ChIP-Seq, and mH2A1 binding inhibited EGFR/CDK5-mediated STAT3-targeted gene expression. Consistent with a previous study[57], degradation of mH2A1 proteins may form a more accessible chromosome structure for STAT3 and thereby promotes STAT3 binding with its target gene promoters, resulting in enhanced STAT3 signaling in GBM. Our study not only advances our current knowledge of the regulation of macroH2A1 ubiquitination and stability but also reveals new functions of macroH2A1 in glioma tumorigenesis.

In conclusion, this study identifies a signaling cascade by which CDK5 promotes STAT3 signaling activation and GBM tumorigenicity through the dual roles of nuclear TRIM59. These results provide critical insights in advancing our understanding of CDK5 functions in cancer. Given the clinical and functional significance, CDK5/TRIM59 signaling axis may be attractive targets for GBM treatment.

## Methods

**Cell lines and cell culture.** HEK293, GBM LN229, and U87 cells were obtained from ATCC (Manassas, VA, USA). GSC R39, R83, primary GBM6, and GBM15 cells were derived from four clinical GBM tumors in Ren Ji Hospital (Shanghai, China). Glioma cells were maintained in 10% fetal bovine serum/Dulbecco's modified Eagle's medium (DMEM), and GSC cells were cultured in DMEM/F12 supplemented with B27 (1:50), heparin (5 mg/ml), basic fibroblast growth factor (20 ng/ml), and EGF (20 ng/ml) as we previously described[21]. All cell lines were cultured at 37 °C and 5% CO$_2$. All cell lines were recently authenticated using STR DNA fingerprinting at Shanghai Biowing Applied Biotechnology Co., Ltd (Shanghai, China), and mycoplasma infection was detected using LookOut Mycoplasma PCR Detection Kit (Sigma-Aldrich).

**Antibodies and reagents.** The following antibodies were used in this study: an anti-TRIM59 antibody (1:500 for WB, 1:50 for IF, ab69639, Abcam); anti-CDK5 (1:1000, sc-249), anti-p-CDK5(Y15) (1:1000 for WB, 1:200 for IHC, sc-12918), anti-GST (1:1000, sc-138), and anti-PIN1 (1:1000, sc-46660) antibodies (Santa Cruz Biotechnology); an anti-Flag M2 antibody (1:1000, F3165, Sigma-Aldrich); anti-CDK1 (1:1000, #19532-1-AP), anti-CDK2 (1:500, #10122-1-AP), anti-CDK4 (1:2000, #11026-1-AP), anti-importin α5 (1:2000, #18137-1-AP), anti-HA (1:2000 for WB, 1:100 for IF, #66006-1-Ig), anti-β-actin (1:5000, #66009-1-Ig), and anti-lamin B1 (1:5000, #66095-1-Ig) antibodies (Proteintech Group); anti-EGFR (1:1000, #4267), anti-p-STAT3 (Y705) (1:1000 for WB, 1:50 for IHC, 9145), anti-phospho-EGFR (Y1173) (1:1000 for WB, 1:50 for IHC, #4407), anti-STAT3 (1:1000, #9139), anti-Myc (1:1000, #2276), anti-tubulin (1:1000, #2148), anti-His (1:1000, #2365), anti-macroH2A1.1 (1:1000, #12455), anti-macroH2A1.2 (1:1000, #4827), and anti-HA (1:1000, #3724) antibodies (Cell Signaling Technology); an anti-macroH2A1 antibody (#07-219, 1:1000 for WB, 1:50 for IF and IHC, Millipore); anti-CD44 (1:100, MA5-13890), anti-Ki-67 (1:500, MA5-14520) antibodies (Invitrogen Antibodies), and anti-phosphoserine/threonine antibody (1:1000, #612548, BD Biosciences). A rabbit polyclonal anti-phospho-TRIM59$^{S308}$ antibody was raised by a pay-for-service vendor through immunizing animals with a synthetic phospho-peptide corresponding to

residues surrounding S308 of human TRIM59. The antibodies were then affinity purified (Abmart Inc., Shanghai, China). Erlotinib was from LC Laboratories. Roscovitine and CHX were from MedChemExpress. GF109203X was from Enzo Life Science. KN-93, HT-59, and leptomycin B were from Sigma. LY290042, U0126, SP600125, and SU6656 were from EMD Biosciences. SB216763, MG-132, and Juglone were from Selleck Chemicals.

**Plasmids.** TRIM59, CDK5, importin α (α1, α3, α4, α5, α6, and α7), PIN1, mH2A1.1, and mH2A1.2 Complementary DNAs (cDNAs) were amplified by RT-PCR from U87 cells, sequenced, and then subcloned into the pLVX-Puro vector or pcDNA3 (Clontech), respectively. TRIM59-truncated constructs were generated as previously reported, which were gifts from Dr. Shigetsugu Hatakeyama[22]. GST-CDK5 was constructed by cloning the wild-type cDNA into pGEX-4T-1 vector. pcDNA3-Myc-STAT3 was derived from pLEGFP-WT-STAT3, which was a gift from George Stark (Addgene plasmid #71450)[60]. pcDNA3-Flag-TC45 was derived from pEFneo-HA-TC45, which was a gift from Dr. Zhijie Chang[61]. Point mutations were generated using a Site-Directed Mutagenesis Kit (Invitrogen) following the manufacturer's protocol.

**IF staining.** Cells were fixed with 4% paraformaldehyde for 15 min, and then permeabilized with 0.2% Triton X-100 for 10 min. Cells were blocked with 3% bovine serum albumin in phosphate-buffered saline (PBS) for 60 min and incubated for 2 h with the primary antibodies indicated in the related figures. After washing three times with PBS, cells were incubated with secondary antibodies for 1 h. Olympus BX53 and ZEISS LSM710 were used to take pictures.

**IP and WB assays.** IP and WB analyses were performed as we previously described[21]. In brief, cells were lysed in IP lysis buffer (20 mM Tris-HCl, pH 7.5, 150 mM NaCl, 1 mM EDTA, 2 mM Na$_3$VO$_4$, 5 mM NaF, 1% Triton X-100, and protease inhibitor cocktail) at 4 °C for 30 min. The lysates were cleared by centrifugation. Equal amounts of cell lysates were immunoprecipitated with specific antibodies and protein G-agarose beads (Invitrogen). Then, the standard WB was performed. Uncropped original blots for figures and Supplementary Figures were included in the Supplementary Fig. 16 in the Supplementary Information in this paper.

**sgRNA-knockout, shRNA-knockdown, and transfection assays.** Single-guide RNA (sgRNA)- knockout, shRNA-knockdown, and transfection assays were performed as we previously described[21]. sgRNA sequences were designed using the MIT online tool (http://crispr.mit.edu). shRNAs were purchased from GeneChem (Shanghai, China). HEK293 cells were transfected with specified DNA and packaging plasmids. Forty-eight hours after transfection, viruses were collected, concentrated, and transduced into various cells as we previously described[47].

**Purification of recombinant proteins and GST pull-down assay.** Purification of recombinant proteins was performed as we previously described[62]. Briefly, HEK293T cells transduced with (His)$_6$-tagged TRIM59 were lysed, sonicated, and centrifuged. Then, the supernatants were purified using a Ni$^+$-NTA column. GST-CDK5 in the plasmid pGEX-4T-1 was transformed into *Escherichia coli* BL21 and purified using glutathione beads according to the manufacturer's procedures. Pull-down assays were performed by incubating purified GST-CDK5 proteins with the cell extracts from U87/EGFR cells with or without EGF stimulation for 30 min. Then the beads were washed and detected by WB analyses.

**In vitro kinase assay.** Briefly, 500 ng purified recombinant (His)$_6$-TRIM59 WT or S308A protein was incubated with 200 μM recombinant active CDK5/p35 protein (Millipore) in 30-μl reaction buffer at 30 °C for 30 min. The reactions were terminated with sodium dodecyl sulfate-polyacrylamide gel electrophoresis loading buffer.

**In vitro isomerization analysis.** In vitro isomerization analysis was performed as previously described[63]. Briefly, to obtain the pure *cis* peptides, peptides were incubated with α-chymotrypsin at 0 °C for 2 min to completely hydrolyze the *trans*

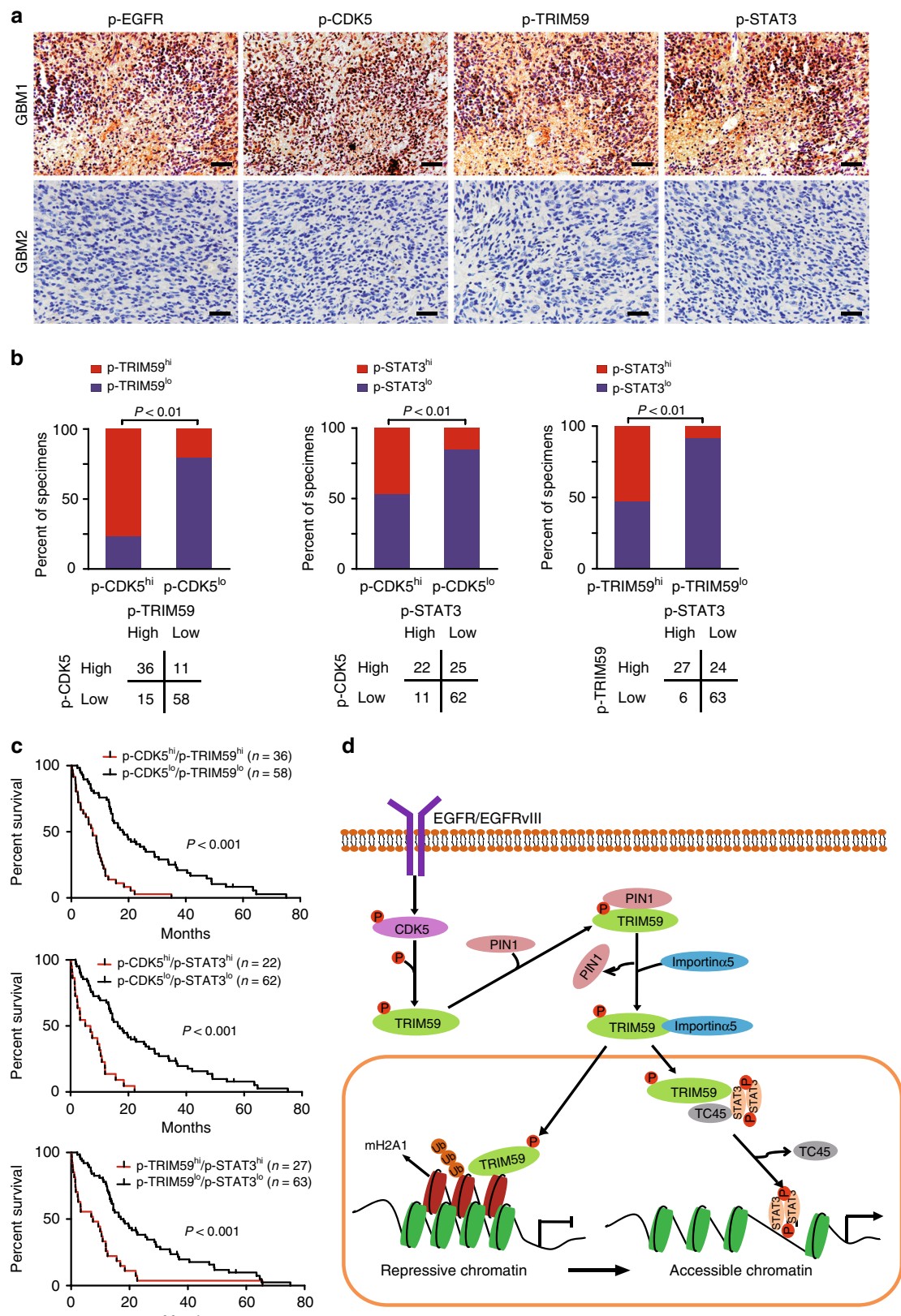

isomer at the 4-nitroanilide bond. After re-equilibration of the pure *cis* peptides, the isomerization was processed. Aliquots were taken at the indicated time, and the released 4-nitroaniline was measured at 405 nm. The *cis* peptide content indicated the isomerization rate.

**RNA isolation and qRT-PCR**. Total RNA was isolated from cells using the Trizol Plus RNA Purification Kit (Thermo Fisher Scientific) and was transcribed into cDNA using the Reverse Transcription Kit (Takara) according to the manufacturer's instructions. qRT-PCR was performed using the Power SYBR Green Master Mix (Life Technologies) in the StepOne Real-Time PCR System (Applied Biosystems, Foster City, USA) with the following primers: for control *ACTB* amplification: 5′-CATGTACGTTGCTATCCAGGC-3′ and 5′-CTCCTTA ATGTCACGCACGAT-3′; for mH2A1 amplification: 5′-AATGCAGCGAGAGA CAACAA-3′ and 5′-CTTCTTCGCTAGCAACTCGG-3′; for *PIM1* amplification:

**Fig. 9** CDK5/TRIM59/STAT3 pathway is clinically prognostic. **a** Representative images of p-EGFR$^{Y1173}$, p-CDK5$^{Y15}$, p-TRIM59$^{S308}$, and p-STAT3$^{Y705}$ IHC in a separate cohort of a total of 120 paraffin-embedded clinical GBM samples. Scale bars, 50 μm. Images are representative of two independent experiments. **b** Correlation of expression between p-CDK5$^{Y15}$, p-TRIM59$^{S308}$, and p-STAT3$^{Y705}$ in **a**. Source data are provided as a Source Data file. **c** Kaplan–Meier survival analysis of GBM patients with tumors expressing indicated proteins. Median survival (in months): p-CDK5$^{hi}$/p-TRIM59$^{hi}$, 7.47; p-CDK5$^{lo}$/p-TRIM59$^{lo}$, 17.97; p-CDK5$^{hi}$/p-STAT3$^{hi}$, 6.00; p-CDK5$^{lo}$/p-STAT3$^{lo}$, 16.33; p-TRIM59$^{hi}$/p-STAT3$^{hi}$, 7.47; p-TRIM59$^{lo}$/p-STAT3$^{lo}$, 17.03. Statistical analysis was performed by the log-rank test. **d** A working model of CDK5-promoted STAT3 signaling activation and GBM tumorigenicity through the dual roles of nuclear TRIM59. EGFR-activated CDK5 phosphorylates TRIM59 at S308, which recruits PIN1 for *cis–trans* isomerization of TRIM59, leading to TRIM59 binding to importin α5 and nuclear translocation. Nuclear TRIM59 enhances STAT3 signaling activation and tumorigenicity not only through maintaining nuclear STAT3 phosphorylation by inhibiting TC45 dephosphorylation but also through promoting macroH2A1 ubiquitination and degradation

---

5′-CGAGCATGACGAAGAGATCAT-3′ and 5′- TCGAAGGTTGGCCTATCTG A-3′. Results were analyzed using the $2^{-(\Delta\Delta Ct)}$ method.

**ChIP-Seq and ChIP-qPCR**. LN229/EGFRvIII cells were treated with vehicle (0.1% dimethyl sulfoxide, erlotinib (10 μM), or Roscovitine (20 μM) for 16 h, and then the cells were cross-linked with formaldehyde to a final concentration of 1%. ChIP samples were prepared as described previously[34]. Immunoprecipitations were performed using 1 μg anti-macroH2A1 antibody (07-219, Millipore) or the relevant non-specific IgG (sc-69831, Santa Cruz Biotechnology). ChIP-Seq libraries generation were performed according to standard protocols using BioScientific DNA Sample Kit. Libraries were sequenced using Illumina HiSeqX Ten platforms. Sequence reads were aligned to the Human Reference Genome (hg19). ChIP-qPCR was performed using a Chromatin Immunoprecipitation Kit (Millipore-Upstate). Briefly, DNAs were immunoprecipitated according to the manufacturer's instructions, and then purified DNAs were measured using qPCR with the following primers for *PIM1* locus: 5′-TCCGGTCCCGGGCCAAGAAT-3′ and 5′-GA GGGCGGGAGGGAGCAGGAA-3′; *MUC1* locus: 5′-TCTTATTTCTCGGCCGCTC TGCTT-3′ and 5′-TGGGTAGGGTACAAGGGCTCTAAT-3′; *AKT3* locus: 5′-TT TGGGAGGCTGAGGCGGGTGGAT-3′ and 5′-ACTACAGGTGTGCGCCACCA CGCC-3′.

**In vivo and in vitro ubiquitination assay**. In vivo ubiquitination assay was performed as we previously described[64]. Briefly, cells were co-transfected with the indicated plasmids for 48 h and then treated with MG-132 for 6 h before harvest. Cells were lysed by the denaturing buffer (6 M guanidine-HCl, 100 mM Na$_2$HPO$_4$/NaH$_2$PO$_4$, and 10 mM imidazole), incubated with nickel beads for 3 h, and analyzed by WB analysis. For in vitro ubiquitination assays, Flag-mH2A1.1 and HA-TRIM59 WT or ΔR mutant proteins purified from the 293T cells were incubated at 37 °C for 2 h in a reaction buffer and then analyzed by WB analysis.

**Limited dilution analysis**. Limiting dilution analysis was performed as described previously[65]. Briefly, dissociated GSC R39 or R83 cells were seeded in 96-well plates at density of 1, 5, 10, 20, 30, 40, or 50. After 2–3 weeks, each well was examined for the formation of tumor spheres. Stem cell frequency was calculated using extreme limiting dilution analysis (http://bioinf.wehi.edu.au/software/elda/).

**Tumorigenesis studies**. Athymic nu/nu female mice aged 6–8 weeks (SLAC, Shanghai, China) were used. Mice were randomly divided into five per group. In total, GSCs ($2 \times 10^5$) transduced with a luciferase reporter were stereotactically implanted into the mouse brain as previously described[64]. Bioluminescence imaging was performed after injection of D-luciferin using the IVIS Lumina imaging station (Caliper Life Sciences). Animal testing and research were complied with all relevant ethical regulations.

**IHC of human glioma specimens**. In accordance to a protocol approved by Shanghai Jiao Tong University Institutional Clinical Care and Use Committee, according to the Declaration of Helsinki, clinical brain tissue specimens were collected at Ren Ji Hospital, School of Medicine, Shanghai Jiao Tong University (Shanghai, China). The investigators obtained informed written consent from the patients. These specimens were examined and diagnosed by pathologists at Ren Ji Hospital. The tissue sections from paraffin-embedded de-identified human GBM specimens were stained with against p-EGFR$^{Y1173}$ (1:50), p-CDK5$^{Y15}$ (1:200), p-TRIM59$^{S308}$ (1:50), and p-STAT3$^{Y705}$ (1:50) antibodies. Non-specific IgGs were used as negative controls. IHC staining was scored as 0–7 according to the percentage of positive cells, as we previously described[21]. Tumors with 0 or 2 staining scores were considered as low expressing and those with scores of 3–7 were considered high expressing.

**Statistics**. GraphPad Prism version 5.0 for Windows (GraphPad Software Inc., San Diego, CA, USA) was used to perform one-way analysis of variance with Newman–Keuls post hoc test or an unpaired, two-tailed Student's *t* test. Kaplan–Meier survival analysis was carried out using log-rank tests. A *P* value of <0.05 was considered significant.

**Study approval**. All animal experiments were performed in accordance with a protocol approved by Shanghai Jiao Tong University Institutional Animal Care and Use Committee (IACUC)-approved protocols, according to National Institutes of Health (NIH) guidelines. All the work related to human tissues and GBM patient specimens for deriving GSCs and primary GBM cells was performed at the Shanghai Jiao University under an Institutional Review Board-approved protocol, according to NIH guidelines, and informed consent was obtained.

## Data availability

ChIP-Seq data reported in this study have been deposited with the Gene Expression Omnibus under the accession GEO ID: GSE130949. The data supporting the finding of this study are available within the article and its Supplementary Information files or available from the corresponding author on reasonable request. The source data underlying Figs. 3e, 6b, d, f, 7e, g, h, j, k, 8b, c, e, f, h, k, 9b and c, and Supplementary Figs. 2b,10a, d, f, 14a and b are provided as a Source Data file.

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

## Acknowledgements

This work was supported in part by National Natural Science Foundation of China (81572467, 81874078 to H.F.; 81470315, 81772663 to Y.L.); the Program for Professor of Special Appointment (Eastern Scholar) at Shanghai Institutions of Higher Learning (2014024), Shanghai Municipal Education Commission-Gaofeng Clinical Medicine Grant Support (No. 20161310), New Hundred Talent Program (Outstanding Academic Leader) at Shanghai Municipal Health Bureau (2017BR021), and the State Key Laboratory of Oncogenes and Related Genes in China (91-17-25) to H.F.; the Doctoral Innovation Fund Projects from Shanghai Jiao Tong University School of Medicine (BXJ201819) to Y.S.; and fund from Jean & Lou Malnati Brain Tumor Institute, Northwestern University Feinberg School of Medicine (Chicago, IL, USA) to S.-Y.C. and B.H.

## Author contributions

Conceptualization: H.F.; investigation: Y.S., Y.L., Y.Z., B.Y. and W.Z.; analysis: Y.S., Y.L. and H.F.; writing/reviewing and editing: Y.S., Y.L., A.A.A., B.H., S.-Y.C. and H.F.; funding acquisition: Y.L. and H.F.

## Additional information

**Competing interests:** The authors declare no competing interests.

