## [Peer Review File · Nature Communications]

Editorial note: Reviewer #3 was unable to comment on the revised version of this manuscript, therefore we recruited Reviewer #4, with similar expertise, to comment on the author's response to Reviewer #3.

Reviewers' comments:

Reviewer #1 (Remarks to the Author): Expert in nuclear import

In this article Youzhou Sang and colleagues investigate the mechanism of TRIM59 mediated tumorigenesis. The authors use a glioblastoma model and several kinase inhibitors to show that TRIM59 is phosphorylated by CDK5 that promotes its nuclear retention. Once in the nucleus, TRIM59 exerts its action through binding and degradation of macroH2A. The authors also show the role of STAT3 in mediating tumorigenesis downstream of TRIM59. The experiments include the use of stem cell model and patient tissue. The authors also propose that CDK5/TRIM59/STAT3 pathway could have prognostic significance.

Specific comments:

Does TRIM59 have a nuclear export signal sequence? Figure 1A and B can be supported by addition of nuclear export inhibitors to show retention.

Figure 7 data is not conclusive. The animal tumor studies over entire treatment period should be shown

Figure 7 results can be strengthened by the addition of IHC analysis and western blot and PCR analysis on animal tumor tissue

Figure 8 Working model legends should be enhanced

Minor comments:

Some typos were observed that need to be checked and corrected

Reviewer #2 (Remarks to the Author): Expert in cancer and ubiquitin

The authors seek to understand the molecular mechanism underlying how CDK5-mediated phosphorylation of TRIM59 promote its nuclear translocation to trigger TRIM59-mediated polyubiquitination and degradation of macroH2A1 to facilitate glioblastoma tumor progression. The paper is clearly written, however, the following concerns should be addressed before its publication at Nature Communications.

1. Figure 1B and S1B, it is clear that EGF not only promotes TRIM59 nuclear localization but also elevates the protein abundance of TRIM59. The authors should examine whether EGF affects TRIM59 mRNA levels or its protein stability. If protein stability of TRIM59 is affected, the authors should examine whether the S308D becomes more stable while S308A is destabilized and what is the possible underlying mechanism?
2. Figure 1C and 1E, Roscovitine is not a specific CDK5 inhibitor which might inhibit many relevant CDKs including CDK1, CDK2 and CDK4. Have the authors excluded the possibility that these CDKs might also involve in regulating TRIM59? It will be important for the authors to compare side by side as they did in Figure 1E the effects of knocking down CDK1, CDK2 and CDK4 vs CDK5 towards TRIM59 nuclear localization.
3. Figure 1D, it will be nice to include TRIM59 IB in whole cell lysates panel.
4. Figure 2B, the authors should provide the possible mechanism of why EGF triggers TRIM59 interaction with CDK5. Is it due to elevated TRIM59 expression or activation of CDK5 kinase activity. Will TRIM59 bind to WT-CDK5 or Kinase dead-CDK5 with comparable affinity?
5. Figure 2D, it is clear that the c-terminus of TRIM59 might mediate the interaction with CDK5. It will be important for the authors to further demonstrate whether the truncation mutant of aa 280-aa 403 (termed D5) is able to interact with CDK5 as the D1 mutant does.
6. Figure 2G, have the authors detected S308 phosphorylation by Mass Spectrometry?
7. Figure S2A, the authors should show whether these sites are evolutionarily conserved.
8. Figure 3F, how will Pin1 pharmaceutical inhibitor affect the nuclear location of TRIM59 WT, vs TRIM59 S308A or S308D mutant?
9. Figure 4A, is this NLS sequence evolutionarily conserved?

10. Figure 5D, does expression of TRIM59 affect macroH2A1 mRNA levels or shorten its half-life?
11. Figure 6, will shPin1 or shimportin5 also elevate macroH2A1 as shTRIM59 does?
12. Figure 6G-H, have the authors determined the ubiquitinated Lysine residues and whether mutating these Lys to Arg will abolish TRIM59-mediated degradation of macroH2A1?
13. Figure 6G, it will be very important to show whether in cells expressing WT-TRIM59, inhibiting CDK5 can reduce macroH2A1 ubiquitination, while in cells expressing S308D mutant form of TRIM59, inhibiting CDK5 cannot reduce macroH2A1 ubiquitination.

Reviewer #3 (Remarks to the Author): Expert in GBM and signalling

In this study titled "CDK5-dependent phosphorylation and nuclear translocation of TRIM59 promotes macroH2A1 ubiquitination and tumorigenicity", Sang et al. propose CDK5/TRIM59 axis as a therapeutic target acting downstream of EGFR signaling in GBM. The authors provided a set of evidence that nuclear translocation of TRIM59 as a means to dysregulate tumor-suppressive macroH2A1 functions to bring about pro-tumor effects of EGFR signaling.

This study addresses issues of high interest in the field of non-canonical histone-mediated changes in cell signaling and functions and provides convincing mechanistic evidence regarding the interactions of the atypical cyclin-dependent kinase with TRIM59 and its subsequent commitment to nuclear compartmentalization. This study nicely takes off from an earlier work (Sang et al., Cancer Research 2018) and adds further dimensions to the purported nuclear effects of TRIM59, a primarily cytosolic protein. In the first half of the study, the authors show that CDK5 physically interacts with TRIM59 and this association is abrogated by the lack of the C-terminal domain of TRIM59. Furthermore, the authors offer experimental validation for putative phosphorylation sites for TRIM59 and its functional consequences. They also show the specific importance of PIN1 in relation to importin α 5-mediated nuclear export of TRIM59. The latter part of the study deals with the downstream effects of this phenomenon and identify macroH2A1 ubiquitin-mediated degradation as the precipitating event to promote the pro-tumorigenic effects of TRIM59 and thus EGFR signaling.

The merits of this study lie in the extensive use biochemical manipulations and characterization of the pathway and supporting clinical evidence therein. However, there are several issues in the manuscript that need to be addressed.

Major concerns:

- Given the continuous failure of the EGFR targeting therapeutics for GBM, it is not clear how this novel molecular signaling provides clinically relevant advance in GBM therapeutics.
- The study will greatly benefit from ChIP-qPCR or ChIPSeq characterization of macroH2A1 binding on relevant GBM genes following abrogation of CDK5-TRIM59 axis with roscovitine and/or erlotinib. What are the variations between sites that are affected by the roscovitine and erlotinib and those that are unaffected? A more comprehensive idea about particular TF sites and comparisons to known maps would be of potential interest.
- The authors need to clarify if the ectopic re-expression of TRIM, PIN and CDK5 mutants were performed in their respective knockout cells in the immunostaining experiments and need to provide appropriate controls (sh, sg harboring non-treated cells) and quantification.
- Further immunohistochemical characterization of molecules of interest and additional tumor markers of stemness, growth in PDX xenograft model (Fig. 7G) would shed light on the nature of the tumor growth/ regression.
- Immunofluorescence images for CDK5 inhibitor study need to be more robust to support the western blot results.

Minor issues:

-The authors need to reduce redundancy in depicting results (Fig 1C and Fig 1D lower left) and should resort to intuitive grouping of data figures (especially for Figs. 6 C, D, E, F).

Point-to-point response to the reviewers' comments

Reviewer #1 (Expert in nuclear import):

1. Does TRIM59 have a nuclear export signal sequence? Figure 1A and B can be supported by addition of nuclear export inhibitors to show retention.

Response: We appreciate the reviewer for this excellent suggestion. As suggested, we have performed *in silico* analysis through NetNES 1.1 Server (<http://www.cbs.dtu.dk/services/NetNES/>) and identified a nuclear export signal sequence (233-LELMALTISLQEE-245) in TRIM59. Then, we performed new experiments and found that pre-treatment with leptomycin B (LMB) inhibitor of nuclear export retained EGF-induced TRIM59 nuclear localization which was inhibited by erlotinib (Fig. 1a and 1b). This has been described in page 6.

2. Figure 7 data is not conclusive. The animal tumor studies over entire treatment period should be shown.

Response: As required by the reviewer, we have performed a new set of animal experiments. We included the bioluminescence images of the animal tumor study during the entire treatment in the revised Fig. 8g. The original Figure 7 is now the Figure 8 in the revised manuscript.

3. Figure 7 results can be strengthened by the addition of IHC analysis and western blot and PCR analysis on animal tumor tissue.

Response: As suggested by the reviewer, we have performed the suggested experiments and included the new data of IHC analysis, Western blotting, and PCR analysis on animal tumor tissues in the revised Fig. 8i to 8k. This result has been described in page 20.

4. Figure 8 Working model 8 legends should be enhanced.

Response: We appreciate the reviewer to point out this. We have rewritten our description for the working model in the Figure 9 (was Figure 8) in the revised manuscript..

5. Some typos were observed that need to be checked and corrected.

Response: We apologize for the typos and errors. We have carefully examined the entire manuscript and corrected all of grammatical errors. The revised manuscript has been thoroughly edited by Professor Shi-Yuan Cheng and a native English speaker, Dr. Angel A. Alvarez. Both of them have been co-authors in this manuscript. Drs. Cheng and Alvarez are faculty members at the Northwestern University in Chicago, USA.

Reviewer #2 (Expert in cancer and ubiquitin):

1. Figure 1B and S1B, it is clear that EGF not only promotes TRIM59 nuclear localization but also elevates the protein abundance of TRIM59. The authors should examine whether EGF affects TRIM59 mRNA levels or its protein stability. If protein stability of TRIM59 is affected, the authors should examine whether the S308D becomes more stable while S308A is destabilized and what is

the possible underlying mechanism?

Response: We appreciate this excellent comment by the reviewer. As recently we described [Cancer Res. 2018, 78(7): 1792-1804], TRIM59 was transcriptionally upregulated by EGFR/EGFRvIII in gliomas. We also performed new experiments and found that TRIM59 degradation was not affected by EGF treatment compared with the control (Supplementary Fig. 2a and 2b). This has been described in page 6.

2. Figure 1C and 1E, Roscovitine is not a specific CDK5 inhibitor which might inhibit many relevant CDKs including CDK1, CDK2 and CDK4. Have the authors excluded the possibility that these CDKs might also involve in regulating TRIM59? It will be important for the authors to compare side by side as they did in Figure 1E the effects of knocking down CDK1, CDK2 and CDK4 vs CDK5 towards TRIM59 nuclear localization.

Response: We appreciate this critical comment. To exclude possibilities that other relevant CDKs including CDK1, CDK2, and CDK4 might also involve in CDK-mediated TRIM59 nuclear translocation, we knocked out CDK1, CDK2, CDK4, as well as CDK5 in LN229/EGFR cells and assessed EGF-induction of TRIM59 nuclear translocation. As shown in Supplementary Fig. 3, only CDK5 knockout but not other CDK knockout markedly reduced EGF-induced TRIM59 nuclear translocation. This has been described in page 7.

3. Figure 1D, it will be nice to include TRIM59 IB in whole cell lysates panel.

Response: We appreciate the reviewer to point out this. We have added WB data for TRIM59 protein in whole cell lysates in the revised Fig. 1e.

4. Figure 2B, the authors should provide the possible mechanism of why EGF triggers TRIM59 interaction with CDK5. Is it due to elevated TRIM59 expression or activation of CDK5 kinase activity. Will TRIM59 bind to WT-CDK5 or Kinase dead-CDK5 with comparable affinity?

Response: As commented by the reviewer, we have performed the suggested experiments and found that ectopic expression of TRIM59 enhanced its association with CDK5 in a dose-dependent manner (Supplementary Fig. 4a). We also investigated the association of CDK5 WT or KD with TRIM59 WT, S308A, or S308D, and found that CDK5 KD or TRIM59 S308A mutant had reduced its ability to interact with their corresponding WT partner, CDK5 or TRIM59, respectively, suggesting that both CDK5 kinase activity and p-S308 of TRIM59 are critical for CDK5-TRIM59 interaction (Supplementary Fig. 4b). This has been described in page 8.

5. Figure 2D, it is clear that the c-terminus of TRIM59 might mediate the interaction with CDK5. It will be important for the authors to further demonstrate whether the truncation mutant of aa 280-aa 403 (termed D5) is able to interact with CDK5 as the D1 mutant does.

Response: As commented by the reviewer, we have performed the suggested experiments by including D5. Our new data showed that both D1 and D5 but not other three deleting mutant of TRIM59 were able to associate with CDK5 in LN229/EGFR cells with EGF stimulation. We included this new data in the revised Fig. 2e. This has been described in page 8.

6. Figure 2G, have the authors detected S308 phosphorylation by Mass Spectrometry?

Response: The S308 residue of TRIM59 was identified to be phosphorylated by large-scale quantitative proteomics in ovarian tumors (Mol Cell Proteomics 2014,13:1690-704). We did not perform the Mass Spectrometry for detecting p-S308 in TRIM59 in this study. However, our comprehensive biochemical studies, in particular our specific anti-phospho-S308-TRIM59 data provide compelling evidence that the S308 residue is phosphorylated by CDK5 in glioma cells with EGF stimulation.

7. Figure S2A, the authors should show whether these sites are evolutionarily conserved.

Response: As commented by the reviewer, we have performed the suggested analysis. We found that S308 is highly conserved in TRIM59 among various species. This data is included in the revised Supplementary Fig. 5d. This has been described in page 9.

8. Figure 3F, how will Pin1 pharmaceutical inhibitor affect the nuclear location of TRIM59 WT, vs TRIM59 S308A or S308D mutant?

Response: As commented by the reviewer, we have performed the suggested experiments. We found that nuclear translocation of TRIM59 WT or S308D mutant was markedly reduced in LN229/EGFRvIII cells treated with Juglone inhibitor for PIN1. This data is included as a new Supplementary Fig. 6 in the revised manuscript. This has been described in page 11.

9. Figure 4A, is this NLS sequence evolutionarily conserved?

Response: As commented by the reviewer, we have performed the suggested analysis. Our data showed that the NLS sequence is conserved among various species. This data is included as a new Fig. 4a in the revised manuscript. This has been described in page 12.

10. Figure 5D, does expression of TRIM59 affect macroH2A1 mRNA levels or shorten its half-life?

Response: As commented by the reviewer, we have performed the suggested experiments and found that expression of exogenous TRIM59 decreased the levels of mH2A1 protein but not H2A1 mRNAs (Supplementary Fig. 10a and 10b). Moreover, TRIM59 overexpression markedly promoted the degradation of endogenous mH2A1 (Supplementary Fig. 10c to 10f). This has been described in page 15.

11. Figure 6, will shPin1 or shimportin5 also elevate macroH2A1 as shTRIM59 does?

Response: Given the critical role of PIN1 and importin α 5 in TRIM59 nuclear translocation, we separately knocked down PIN1 or importin α 5. As expected, levels of mH2A1 protein were increased after knockdown of PIN1 or importin α 5 (Supplementary Fig. 9a and 9b). On the other hand, overexpression of TRIM59 decreased the levels of mH2A1 protein but not its mRNAs (Supplementary Fig. 10a and 10b). This has been described in page 15.

12. Figure 6G-H, have the authors determined the ubiquitinated Lysine residues and whether

mutating these Lys to Arg will abolish TRIM59-mediated degradation of macroH2A1?

Response: As suggested by the reviewer, to identify the ubiquitination sites of mH2A1, we first analyzed the published large-scale quantitative proteomics data and found that multiple lysine (K) residues (K18, K116, K123, K167, K189, K235, K251, K285, K292, K295) of mH2A1 were ubiquitinated in that study [Mertins *et al.*, *Nat Methods* **10**, 634-637 (2013); Kim *et al.*, *Mol Cell* **44**, 325-340 (2011)]. We further mutated each lysine (K) to arginine (R) in mH2A1 and revealed that only mH2A1 K167R mutant, but not other KR mutants, markedly reduced mH2A1 ubiquitination regulated by TRIM59 (Supplementary Fig. 12). This has been described in page 16.

13. Figure 6G, it will be very important to show whether in cells expression WT-TRIM59, inhibiting CDK5 can reduce macroH2A1 ubiquitination, while in cells expressing S308D mutant form of TRIM59, inhibiting CDK5 cannot reduce macroH2A1 ubiquitination.

Response: We thank the reviser for the excellent comment. We performed the suggested experiments. Our data shows that Roscovitine treatment decreased mH2A1 ubiquitination regulated by TRIM59 WT but not S308D mutant. This data is included in the revised Fig. 6h in the revised manuscript.

Reviewer #3 (Expert in GBM and signalling):

1. Given the continuous failure of the EGFR targeting therapeutics for GBM, it is not clear how this novel molecular signaling provides clinically relevant advance in GBM therapeutics.

Response: We appreciate this critical comment by the reviewer. The prevalence of EGFR as an oncogenic driver across many prominent types of human cancers including GBM renders EGFR an appealing target for therapeutic intervention. However, a number of mechanisms have been proposed to underlie GBM resistance to EGFR-targeted therapies. One of such mechanisms is that EGFR inhibition induces a rapid adaptive response that mediates resistance to EGFR inhibition (Guo *et al.*, *Nat Neurosci* **20**, 1074-1084 (2017)). On the other hand, it has been recognized that EGFR signaling network in GBM is highly heterogeneous (Furnari, FB *et al.*, 2015, *Nat Rev Can.*, 15:302-315). Our findings in this study provide novel insights of the mechanisms underlying EGFR-driven GBM tumorigenic phenotype. This new knowledge will be highly useful for developing effective treatments for EGFR expressing GBM using a combination inhibition of EGFR and the newly described CDK5/TRIM59 axis. We have included this view in our revised discussion section on page 22.

2. The study will greatly benefit from ChIP-qPCR or ChIPSeq characterization of macroH2A1 binding on relevant GBM genes following abrogation of CDK5-TRIM59 axis with roscovitine and/or erlotinib. What are the variations between sites that are affected by the roscovitine and erlotinib and those that are unaffected? A more comprehensive idea about particular TF sites and comparisons to known maps would be of potential interest.

Response: We thank the reviewer for this insightful comment. As suggested by the reviewer, we performed genome-wide mapping of mH2A1 binding using ChIP-Seq in LN229/EGFRvIII cells treated with or without EGFR inhibitor erlotinib or CDK5 inhibitor Roscovitine (Fig. 7a). Our data

showed that compared with the vehicle treatment cells, mH2A1 bindings in the genomic DNA showed an increase in LN229/EGFRvIII cells treated with erlotinib or Roscovitine (Fig. 7b). We then performed Gene Ontology analysis and found that both treatments of erlotinib and Roscovitine markedly influenced mH2A1 binding at genes from many pathways (Supplementary Table 1). The top five EGFR-driven pathways, including JAK-STAT signaling pathway, were shown (Fig. 7c). Integrative Genomics Viewer (IGV) snapshots of ChIP-seq tracks further revealed significant increases in mH2A1 binding at the STAT3-target genes, *PIM1*, *MUC1*, and *AKT3* following erlotinib or Roscovitine treatment (Fig. 7d). Together, our data corroborate with the reviewer's comment and strongly support our observation that mH2A1 binding is mediated by EGFR and CDK5 activity, and mH2A1 may regulate STAT3 signaling. These data are included as new Figure 7a to 7e in the revised manuscript.

3. The authors need to clarify if the ectopic re-expression of TRIM, PIN and CDK5 mutants were performed in their respective knockout cells in the immunostaining experiments and need to provide appropriate controls (sh, sq harboring non-treated cells) and quantification.

Response: We appreciate the reviewer for this excellent comment and apologize for the confusion. In our data in Fig. 2j and Fig. 4b, we only overexpressed HA-TRIM59 WT and mutants in LN229/EGFR cells. We have revised our description for this data on pages 10 and 12. We trust that our revised wording can clarify the confusion.

4. Further immunohistochemical characterization of molecules of interest and additional tumor markers of stemness, growth in PDX xenograft model (Fig. 7G) would shed light on the nature of the tumor growth/ regression.

Response: As commented by the reviewer, we have performed the suggested experiments. Our data show that compared with the shRNA control, TRIM59 knockdown decreased expression of p-STAT3, proliferation marker Ki-67 and cancer cell stemness marker CD44, whereas the levels of mH2A1 was elevated in GSC xenograft tumor tissues by immunohistochemical (IHC) analyses. Re-expression of shRNA-resistant TRIM59 WT or S308D but not S308A mutant in TRIM59 knockdown cells rescued the levels of p-STAT3, Ki-67, and CD44 expression while reducing mH2A1 levels (Fig. 8i). Additionally, immunoblotting (IB) and qRT-PCR further confirmed the above results (Fig. 8j and 8k). These data are included in the revised Figure 8 panels in the revised manuscript,

5. Immunofluorescence images for CDK5 inhibitor study need to be more robust to support the western blot results.

Response: We apologize for the low quality of this data. We have re-performed this sets of experiments and replaced the original images with new images in high quality in the revised Fig.1d.

6. The authors need to reduce redundancy in depicting results (Fig 1C and Fig 1D lower left) and

should resort to intuitive grouping of data figures (especially for Figs. 6 C, D, E, F).

Response: We appreciate the reviewer for this comment. We have revised the description for the results in Fig.1 and re-grouped the data in the revised Fig. 6.

REVIEWERS' COMMENTS:

Reviewer #1 (Remarks to the Author):

Comments have been addressed. The manuscript is now technically sound. The results support the overall conclusions.

Reviewer #2 (Remarks to the Author):

The authors have addressed most of the raised concerns during this round of revision.

Reviewer #4 (Remarks to the Author):

In this manuscript, the authors formed a very good work. If the authors use primary culture GBM cells to confirm their work, the results will be much more solid. Overall, I consider this work meet the quality of Nature Communication.

Point-to-point response to the reviewers' comments

Reviewer #4:

1. *In this manuscript, the authors formed a very good work. If the authors use primary culture GBM cells to confirm their work, the results will be much more solid. Overall, I consider this work meet the quality of Nature Communcation.*

Response: We appreciate the reviewer for this excellent suggestion. As required by the reviewer, we have performed a new set of experiments using patient-derived primary GBM6 and GBM15 cells with high levels of endogenous EGFR WT (Supplementary Fig. 14a) and found that treatment of erlotinib and Roscovitine significantly inhibited EGFR phosphorylation (p-EGFR) and/or p-CDK5 in GBM6 and GBM15 cells, respectively (Supplementary Fig. 14a). Both erlotinib and Roscovitine treatments markedly inhibited TRIM59 nuclear translocation (Supplementary Fig. 14a), p-STAT3 (Supplementary Fig. 14a), and *PIM1* mRNA expression (Supplementary Fig. 14b), but increased mH2A1 protein levels (Supplementary Fig. 14a), validating our observations in LN229/EGFR GBM, GSC R39, and R83 cells.

We also re-expressed different shRNA-resistant TRIM59 constructs, WT, the non-phosphorylatable S308A, or the phosphor-mimic S308D mutant in GBM6 and GBM15 cells with a TRIM59 shRNA. Compared with the shRNA control, knockdown of TRIM59 significantly inhibited p-STAT3 (Supplementary Fig. 14c) and *PIM1* mRNA expression (Supplementary Fig. 14d), but increased mH2A1 protein levels (Supplementary Fig. 14c). Re-expression of shRNA-resistant TRIM59 WT or S308D mutant rescued p-STAT3 (Supplementary Fig. 14c), *PIM1* mRNA expression (Fig. 8e and Supplementary Fig. 14d), but inhibited mH2A1 protein levels (Fig. 8d and Supplementary Fig. 14c).